# Graph Alignment for Benchmarking Graph Neural Networks and Learning Positional Encodings

Adrien Lagesse [1]   Marc Lelarge [1]

## Abstract

We propose a novel benchmarking methodology for graph neural networks (GNNs) based on the graph alignment problem, a combinatorial optimization task that generalizes graph isomorphism by aligning two unlabeled graphs to maximize overlapping edges. We frame this problem as a self-supervised learning task and present several methods to generate graph alignment datasets using synthetic random graphs and real-world graph datasets from multiple domains. For a given graph dataset, we generate a family of graph alignment datasets with increasing difficulty, allowing us to rank the performance of various architectures. Our experiments prove that there is an optimal task difficulty for having a statistically relevant ranking of different models and that, even on a structure-only task, anisotropic models perform better compared to isotropic ones. To further prove that our synthetic task capture meaningful information, we show its effectiveness for self-supervised GNN pre-training: the learned node embeddings can be leveraged as positional encodings by transformers for graph regression or can be used to reconstruct the full structure of the graph with 98% accuracy. To support reproducibility and further research, we provide an open-source Python package to generate graph alignment datasets and benchmark new GNN architectures. The source code is available at *graph-alignment-benchmark*.

## 1. Introduction

Graphs are versatile and complex mathematical structures, making them adaptable for representing various data types across many scientific fields. In recent years, multiple architectures of Graph Neural Networks (GNNs) were developed (Kipf & Welling, 2017; Veličković et al., 2018; Togninalli et al., 2019; Brody et al., 2022), enabling the application of machine learning and deep learning techniques to graph-based data. These models are used to tackle challenges in areas such as chemistry (Gilmer et al., 2017; Duvenaud et al., 2015; Reiser et al., 2022), material science (Jain et al., 2013; Reiser et al., 2022; Chanussot et al., 2021; Tran et al., 2023), biology (Stokes et al., 2020; Zhang et al., 2021), and social networks (Fan et al., 2019; Ying et al., 2018). In parallel, several graph datasets were compiled to create benchmarks that encompass a diverse set of tasks (Hu et al., 2020; Morris et al., 2020; Dwivedi et al., 2022), facilitating the comparison and evaluation of different architectures. Moreover, task-specific datasets are often employed to compare architectures designed for particular applications, such as those by Chanussot et al. (2021); Tran et al. (2023); Lee et al. (2023) for material science, Wu et al. (2017) for chemistry, and Li et al. (2024) for the SAT problem.

However, unlike text, images, videos, audio, and time-series data, a graph is not an Euclidean data type (Bronstein et al., 2021). Each graph in a dataset has a unique topology that carries critical information that needs to be interpreted by the GNN. This creates a dual challenge in real-world datasets, where both the structural information of the graph and its features (node features, edge features, graph features, *etc...*) must be accounted for (Liu et al., 2022). While leveraging all available information is valuable in practice, it is not essential for a benchmark. In fact, it often complicates the process, as assessing an architecture's structural and feature understanding simultaneously can be challenging. We risk diluting structural information, leading to concerning results, such as more theoretically powerful architectures underperforming compared to simpler models like the Graph Convolutional Network (Kipf & Welling, 2017). Currently, structural understanding in GNNs is typically assessed either theoretically through the Weisfeiler-Lehman test (Xu et al., 2019; Morris et al., 2019; Azizian & Lelarge, 2021) or experimentally using combinatorial optimization benchmarks that lack key features we describe in this paper.

We believe there is a gap in the existing benchmarks avail-

[1]INRIA, École Normale Supérieure - PSL, Paris, France. Correspondence to: Adrien Lagesse <contact@adrien-lagesse.io>.

*Proceedings of the 43rd International Conference on Machine Learning*, Seoul, South Korea. PMLR 306, 2026. Copyright 2026 by the author(s).

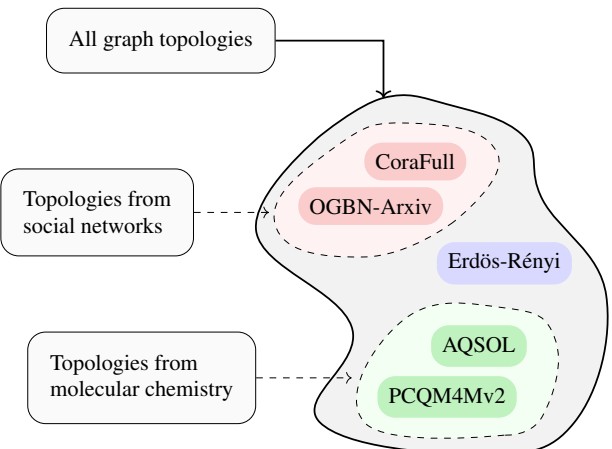

*Figure 1.* Visualization of graph topologies: Graph topologies differ widely across application domains, and the performance of GNN architectures varies accordingly based on these topological differences. Furthermore, different tasks may share the same graph topologies (see Section O).

able to the GNN community to have a more fine-grained metric to compare different models in the same Weisfeiler-Lehman class and to quantify their structural expressivity within that class. Additionally, we argue that evaluating GNN architectures based on graph topology provides broader, more generalizable information compared to task-specific performance. Indeed, this approach is particularly valuable as many different tasks share similar underlying graph topologies (see Figure 1).

**Contributions**

1. We introduce a novel benchmarking task centered on the *Graph Alignment Problem* to evaluate the structural analysis capabilities of GNNs and we show that this task is a generalization of the Weisfeller-Lehman test. We provide a comprehensive methodology for generating datasets using both synthetic and real-world graph data, which also enables the creation of datasets with controllable levels of difficulty, allowing for more nuanced evaluations - Section 4.

2. We further demonstrate the applicability of our framework by benchmarking widely used Message-Passing Neural Network (MPNN) architectures on datasets with varying graph topologies - Section 5.

3. Our empirical analysis highlights that the graph alignment task compress the graph structure and can be effectively used for self-supervised GNN pre-training. The learned node embeddings can serve as positional encodings and can be leveraged by transformer-based models for downstream graph regression tasks and high accuracy graph reconstruction - Section 6.

4. Additionally, we provide an open-source Python package that facilitates the creation of new datasets following our methodology, as well as tools for benchmarking alternative GNN architectures.

## 2. Related Work

**Existing methods for benchmarking GNNs.** Understanding and comparing deep learning models is crucial to advance the field of machine learning, *ImageNet* (Deng et al., 2009) was instrumental in shaping the current computer vision landscape, and advances in other machine learning fields are highly correlated with high-quality benchmarks. To benchmark graph neural networks, general-purpose and widely used datasets were compiled in *TUDataset* (Morris et al., 2019), *Open Graph Benchmark* (Hu et al., 2020) and *Benchmarking Graph Neural Networks* (Dwivedi et al., 2022). In addition to those datasets, specialized benchmarks also exist in structural chemistry (Wu et al., 2017), material-science (Jain et al., 2013; Chanussot et al., 2021; Tran et al., 2023; Lee et al., 2023) as well as benchmarks based on combinatorial optimization tasks such as neural algorithmic reasoning (Veličković et al., 2022) and SAT (Li et al., 2024). Among these datasets, only neural algorithmic reasoning and combinatorial optimization tasks specifically benchmark the structural understanding of a graph neural network. However, they have a major limitation: their inability to adapt to a wide variety of graph topologies. The PATTERN, and CLUSTER datasets from Dwivedi et al. (2022) are inherently based on the Stochastic Block Model (SBM), which can only represent a small portion of graph topologies (molecular graphs for example are not well modeled by the SBM). The CSL dataset from Dwivedi et al. (2022) requires synthetically adding cycles in the graphs, modifying its topology. In other combinatorial tasks such as neural algorithmic reasoning, SAT, or mixed integer programming, the graph topology is strongly correlated to the underlying problem (a more comprehensive comparison between our benchmark and existing combinatorial optimization tasks is provided in Section C.2.1). Nevertheless, combinatorial optimization problems are good candidates for building a benchmark that prioritizes structural understanding, as they are inherently linked only to the unlabeled graph and are a great testbed for GNNs according to Bechler-Speicher et al. (2025).

**Graph Alignment Problem.** A fundamental problem in graph theory is the *Graph Alignment Problem* (or *Graph Matching*). The goal of this task is to align two unlabeled graphs while maximizing the number of common edges, it is a NP-hard problem closely related to the quadratic assignment problem. It is also interesting to note that solving the *Graph Alignment Problem* enables us to solve many other fundamental problems such as the Traveling Salesman Problem (the 1-0 Traveling Salesman Problem, where the

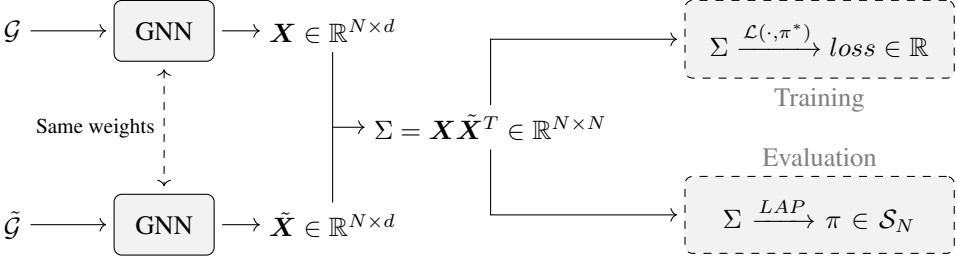

*Figure 2.* The siamese algorithm for predicting a solution to the *Graph Alignment Problem*. During training, we use the binary cross-entropy loss between the predicted similarity matrix $\Sigma$ and a solution $\pi^*$ of the *Graph Alignment Problem*. During evaluation, we use the Hungarian algorithm to extract a permutation from $\Sigma$ by solving the *Linear Assignment Problem*. In practice, using PyTorch Geometric, we represent the graph solely by its *edge_index*, with all node features set to a constant value (e.g., 1) to focus exclusively on the graph structure.

shortest tour is examined on a graph without edge weights), and Maximum Clique Problem. To build our benchmarking task, the *Graph Alignment Problem* has a great property compared to other combinatorial optimization tasks: we can plant a synthetic solution without altering the topology of the graph (see Section 4) as we are working on pairs of graphs. Using machine learning to approximate solutions of the *Graph Alignment Problem* is extensively studied (Cullina & Kiyavash, 2017; Ding et al., 2018; Cullina et al., 2019; Ganassali & Massoulié, 2020; Ganassali et al., 2021; Lelarge, 2025) and in particular, using deep learning and graph neural networks yielded good results (Nowak et al., 2017; Li et al., 2019; Azizian & Lelarge, 2021). However, these papers lack two key components to be adapted in a benchmark, on one hand, they specify fixed graph neural network architectures, and on the other hand, they use a synthetic dataset based on the Erdös-Rényi random graph model, which in practice is rarely found in real-world applications. In addition, Ying et al. (2021b) and He et al. (2024) also consider a similar problem but are more interested in the absolute performance of their algorithm on the alignment task rather than designing a good methodology to compare different GNNs on different graph topologies.

## 3. Siamese architecture for the Graph Alignment task

The benchmarking task we propose and how we build the corresponding dataset is inspired by Nowak et al. (2017). We use a siamese algorithm that can accommodate any GNN architecture. We call a GNN any equivariant trainable model, *i.e.* a model where applying a permutation to the input graph results in a corresponding permutation of the output.

**Notations.** We denote by $\mathcal{G} = (V, E)$ a simple graph with vertex set $V = \{1, \ldots, N\}$ and edge set $E \subset V \times V$. We denote by $\boldsymbol{A} \in \{0,1\}^{N \times N}$ the adjacency matrix of $\mathcal{G}$, i.e. $\boldsymbol{A}_{i,j} = 1$ if $(i, j) \in E$ and $\boldsymbol{A}_{i,j} = 0$ otherwise. We denote by $\mathcal{S}_N$ the set of permutations, as well as the

set of permutation matrices. In particular, for $\pi \in \mathcal{S}_N$, the associated matrix $\boldsymbol{P}$ is such that $\boldsymbol{P}_{i,j} \in \{0,1\}$ and $\boldsymbol{P}_{i,j} = 1$ if and only if $\pi(i) = j$. We call $\mathcal{B}(p)$ the Bernoulli distribution with parameter $0 \leq p \leq 1$.

**Graph Alignment Problem.** Let $\mathcal{G}$ and $\tilde{\mathcal{G}}$, be two graphs with $N$ vertices such that $\boldsymbol{A}$ and $\tilde{\boldsymbol{A}}$ are their associated adjacency matrices. The *Graph Alignment Problem* consists of aligning $\mathcal{G}$ and $\tilde{\mathcal{G}}$ to maximize the number of overlapped edges - see Equation (1). If $\pi^*$ is a solution of the *Graph Alignment Problem* between $\mathcal{G}$ and $\tilde{\mathcal{G}}$, we say that $\pi^*$ aligns $\tilde{\mathcal{G}}$ with $\mathcal{G}$.

$$\pi^* \in \arg\max_{\pi \in \mathcal{S}_N} \sum_{i,j \in \{1,\ldots,N\}} \boldsymbol{A}_{i,j} \tilde{\boldsymbol{A}}_{\pi(i),\pi(j)} \qquad (1)$$

The *Graph Alignment Problem* generalizes the *Graph Isomorphism Problem*: first, two graphs are isomorphic if and only if they are identical after alignment; second, even when they are not isomorphic, the *Graph Alignment Problem* quantifies their similarity.

We say that $\mathcal{D} = \{(\mathcal{G}_i, \tilde{\mathcal{G}}_i, \pi_i)\}_i$ is a graph alignment dataset, if for all $i$, $\pi_i$ is a solution of the *Graph Alignment Problem* between $\mathcal{G}_i$ and $\tilde{\mathcal{G}}_i$.

We also consider the *Linear Assignment Problem* which consist of solving for a reward matrix $\Sigma \in \mathbb{R}^{N \times N}$ the optimization problem

$$\pi^* \in \arg\max_{\pi \in \mathcal{S}_N} \sum_{i \in \{1,\ldots,N\}} \Sigma_{i,\pi(i)} \qquad (2)$$

As in our setup, $\Sigma$ is the similarity matrix, the Linear Assignment Problem will find a permutation maximizing the global similarity.

While the *Graph Alignment Problem* is NP-hard, the *Linear Assignment Problem* is solvable in $\mathcal{O}(N^3)$ using the Hungarian algorithm.

**Siamese Neural Network for Graph Alignment.** Nowak et al. (2017) introduce a siamese neural network to solve the

*Graph Alignment Problem* for a pair of graphs $(\mathcal{G}, \tilde{\mathcal{G}})$. This architecture can in fact be used with any GNN as shown in Figure 2. The siamese module outputs a similarity matrix $\Sigma \in \mathbb{R}^{N \times N}$ and a higher similarity between node $i$ in $\mathcal{G}$ and node $j$ in $\tilde{\mathcal{G}}$ (i.e., $\Sigma_{ij}$) indicates a higher probability of them being aligned. Intuitively, if $\mathcal{G}$ and $\tilde{\mathcal{G}}$ are already aligned, we want $\Sigma_{i,i} = \langle X_i, \tilde{X}_i \rangle$ to be large, and $\Sigma_{i,j} = \langle X_i, \tilde{X}_j \rangle$ to be small when $i \neq j$. Thus, the goal of the siamese architecture is to compute node representations that are mostly invariant to small perturbations of $\mathcal{G}$ while being orthogonal to the representations of other nodes in the graph. In Section C, we give more nuanced interpretations of the task from a representational learning viewpoint.

During training, we have access to the optimal permutation $\pi^*$ between $\mathcal{G}$ and $\tilde{\mathcal{G}}$, hence we can compute the binary cross-entropy loss:

$$\mathcal{L}(\Sigma, \pi^*) = -\sum_{i=1}^{n} \ln \left( \sigma(\Sigma) \right)_{i, \pi^*(i)}$$

where $\sigma$ is the softmax function. This loss encourages large similarity for nodes that should be matched and low similarity for those that should not.

In addition, the following theorem asserts that this loss works well with graphs with symmetries ($Aut(\mathcal{G}) \neq \{\mathbb{I}d\}$):

**Theorem 3.1.** *(Informal) For any permutation $\pi$ and automorphism $\rho \in Aut(\mathcal{G})$ we have:*

$$\mathcal{L}(\Sigma, \pi) = \mathcal{L}(\Sigma, \pi \circ \rho)$$

*See the full theorem and proof in Theorem B.2.*

During evaluation, we extract a permutation from $\Sigma$ by solving the Linear Assignment Problem (see Equation (2)), which maximizes the overall similarity. Accuracy is defined as the percentage of correctly mapped nodes.

**Theorem 3.2.** *(Informal) The architecture presented in this paper preserves 3 key equivariance properties necessary to the Graph Alignment Problem.*

*See the full theorem and proof in Theorem A.1.*

**Correlated Erdös-Rényi Graphs.** Considering that we are in a supervised learning setting, building a dataset $\mathcal{D} = \{(\mathcal{G}_i, \tilde{\mathcal{G}}_i, \pi_i)\}_i$ where $\pi_i$ is a solution of the graph alignment problem between $\mathcal{G}_i$ and $\tilde{\mathcal{G}}_i$ can also be challenging. Given two graphs, extracting an optimal graph alignment permutation $\pi$ cannot be done in polynomial time, hence the only viable option is to create two graphs that we know how to align without needing to compute the solution. Nowak et al. (2017) introduces an algorithm to generate a pair of correlated Erdös-Rényi graphs. This correlation assures us that with a high probability, both graphs are already aligned. However, this method can only be applied

to generate Erdös-Rényi graphs, which is too restrictive to build a meaningful benchmark that can accommodate many different graph topologies. In Section 4, we present a new method to construct a dataset $\mathcal{D}$ from any graph datasets.

# 4. Building the Graph Alignment datasets

In this section, we present algorithms for constructing a dataset suitable for training a siamese neural network (see Section 3) to address the *Graph Alignment Task*. We introduce key modifications to the method proposed by Nowak et al. (2017), enabling the generation of graph alignment dataset from an existing graph dataset. This approach enables benchmarking models across a variety of graph topologies.

**Correlated Graphs.** Let $\mathcal{G}$ be a simple undirected connected graph with $N$ vertices and let $\boldsymbol{A}$ be its adjacency matrix. We will randomly add and remove edges in $\mathcal{G}$ to obtain a new graph $\tilde{\mathcal{G}}$ correlated with $\mathcal{G}$, let $p_{add}$ and $p_{remove}$ be the probabilities of adding and removing an edge. We sample $\boldsymbol{N}^{add}$ a $N \times N$ matrix such that $\boldsymbol{N}_{ii}^{add} = 0$ and $\boldsymbol{N}_{ij}^{add} = \boldsymbol{N}_{ji}^{add} \sim \mathcal{B}(p_{add})$, similarly, we also sample $\boldsymbol{N}^{remove}$. We then build the adjacency matrix of the $\tilde{\mathcal{G}}$ as follows:

$$\tilde{\boldsymbol{A}}_{ij} = \boldsymbol{A}_{ij}(1 - \boldsymbol{N}_{ij}^{remove}) + (1 - \boldsymbol{A}_{ij})\boldsymbol{N}_{ij}^{add}$$

Due to the wide variety of graph topologies we want to accommodate, we propose two ways to choose $p_{add}$ and $p_{remove}$. First, we define a noise level $0 \leq \eta < 1$, which quantifies the correlation between $\mathcal{G}$ and $\tilde{\mathcal{G}}$. Then we define, depending on the characteristic topology of the graphs in the dataset, $p_{add}$ and $p_{remove}$:

- In order to have a stable average degree between $\mathcal{G}$ and $\tilde{\mathcal{G}}$, we fix $p_{remove} = \eta$ and $p_{add} = \frac{\eta|E|}{N(N-1)-|E|}$. On average, we will remove $\eta|E|$ existing edges, and add $\eta|E|$ new edges. $p_{add}$ is not always defined for dense graphs, it is required that $\eta \leq \frac{N(N-1)}{|E|} - 1$.

- When we are working on a graph with a small average degree we want to avoid building a disconnected graph $\tilde{\mathcal{G}}$ (*e.g.* molecular graphs), we fix $p_{remove} = 0$ to avoid this problem. We also fix $p_{add} = \frac{\eta|E|}{N(N-1)-|E|}$ where $|E|$ the number of edges in $\mathcal{G}$. This specific value assures us, that on average, $\eta|E|$ edges will be added.

**Graph Datasets to Graph Alignment Datasets.** Let $\{\mathcal{G}_i\}_i$ be a graph dataset, this dataset contains only the graph structural information, all the node features, edge features, graph features, *etc...* are removed. We call $\{\mathcal{G}_i\}_i$ a base dataset and we show how to generate a graph alignment dataset from any base dataset.

First, we fix the noise level $\eta$. To build the graph alignment dataset $\mathcal{D}$ at the fixed noise level $\eta$ from the base dataset $\{\mathcal{G}_i\}_i$, we build for each graph $\mathcal{G}_i$, the correlated graph $\tilde{\mathcal{G}}_i$. For $\eta$ there is a high probability that $\mathcal{G}_i$ and $\tilde{\mathcal{G}}_i$ are aligned, hence $(\mathcal{G}_i, \tilde{\mathcal{G}}_i, \mathrm{Id})$ is a valid sample to train the siamese architecture. Rather than adding it directly to the dataset, we sample uniformly a random permutation $\pi_i$ and add $(\mathcal{G}_i, \pi_i^{-1} \circ \tilde{\mathcal{G}}_i, \pi_i)$ to $\mathcal{D}$, which is also a valid sample.

**Working with a large graph.** GNNs are used on extremely large graphs (thousands to millions of vertices), yet, the siamese architecture presented in Section 3 requires the computation of the similarity matrix $\Sigma = \boldsymbol{X}\tilde{\boldsymbol{X}}^T$ which can be slow. In practice, training a GNN using the *Graph Alignment Task* with graphs of more than 10000 vertices is not possible in a reasonable amount of time. To tackle this limitation, we sample the large graph into smaller sub-graphs. Let $\{\mathcal{G}^{big}\}$ be a base dataset containing only a very large graph with $N$ vertices, we apply the algorithm described in Algorithm 1 to build a dataset of $k$ smaller graphs with $N'$ vertices, that share the same local topology as $\mathcal{G}^{big}$. We then use this new dataset as the base dataset to build a graph alignment dataset. This algorithm, known as the BFS sampling algorithm or Random Node Neighbor sampling (Leskovec & Faloutsos, 2006), is closely related to the breadth-first search (BFS) algorithm.

BFS sampling not only preserves the local graph topology but also works particularly well with Message-Passing Neural Networks (MPNNs). From the perspective of a single node, MPNNs aggregate information from neighboring nodes layer by layer. Consequently, for a model with $n$ layers, the embedding of a node depends only on its $n$-hop neighborhood. Therefore, if a node is added to $V'$ at least $n$ steps prior to the final BFS sampling step, the MPNN will compute the same embedding on $\mathcal{G}'$ as it would on $\mathcal{G}^{big}$.

**Generated datasets.** We apply this methodology to generate graph alignment datasets from five base datasets. For each base dataset we generate eight graph alignment datasets with noise levels ranging from 4% to 30%. The base datasets tested in this paper are: Erdös-Rényi, AQSOL, PCQM4Mv2, CoraFull and OGBN-Arxiv. In Table E.4, we summarize key details about the graph alignment datasets we have constructed.

## 5. Empirical ranking of 1-WL GNNs

**Experimental Setup.** We evaluate the structural representation capabilities of established MPNNs on the *Graph Alignment Task* using the forty graph alignment datasets built. The varying noise level $\eta$, controls the difficulty of the task. A higher noise level results in weaker correlations between pairs, making the task more challenging.

We train each GNN model on each graph alignment dataset and evaluate it on the validation split of the same dataset. Each experiment was repeated 10 times.

The architectures evaluated include the Graph Convolutional Network (Kipf & Welling, 2017), Graph Isomorphism Network (Xu et al., 2019), Gated Graph ConvNets (Bresson & Laurent, 2017), Graph Attention Transformer (Veličković et al., 2018), and GATv2 (Brody et al., 2022). A summary of these models is provided in Table E.5, with detailed specifications available in the accompanying Python library.

**Baseline.** As an additional point of comparison, we use the Graph Laplacian Positional Encoding Dwivedi & Bresson (2020); Dwivedi et al. (2022) as a baseline. This approach relies on the spectral properties of the graph to generate node embeddings, bypassing the need for training. We select the 64 most informative eigenvectors, ensuring the node embeddings are directly comparable to those generated by the trained GNNs. Notably, this method provides insight into the performance of purely structural embeddings, serving as a training-free benchmark against which the learning-based models can be evaluated. The baseline results are presented in Table F.6; however, they are omitted from Figure 3 due to their poor performance.

**Results.** We compile for each model and each graph alignment dataset the average alignment accuracy over the 10 runs as well as the standard deviation in Table F.6. In addition, to aid interpretation, we report the accuracy gap, defined as the difference between a model's accuracy and that of the worst-performing model at a given noise level. Figure 3 visualizes this metric for all models across three graph alignment datasets based on the results summarized in Table F.6.

**Analysis.** We observe that task difficulty (*i.e.*, noise level) significantly affects the performance gap between benchmarked models. When the task is too easy ($\eta$ small) or too difficult ($\eta$ large), performance differences between models are minimal. However, at an optimal noise level, the accuracy gap is largest, making it easier to distinguish between better and worse models. At this level, error bars do not overlap, providing greater statistical significance. Because of this wide performance gap between the different architectures, the results of our benchmark are less sensitive to the random seed as well as the hyperparameter choice (learning rate, regularization, etc...).

Furthermore, performance at the optimal noise level generalizes well across other noise levels.

- Models with similar performance at the optimal noise level remain comparable across all noise levels.

- A model that performs better at the optimal noise level consistently outperforms or matches others at any noise level.

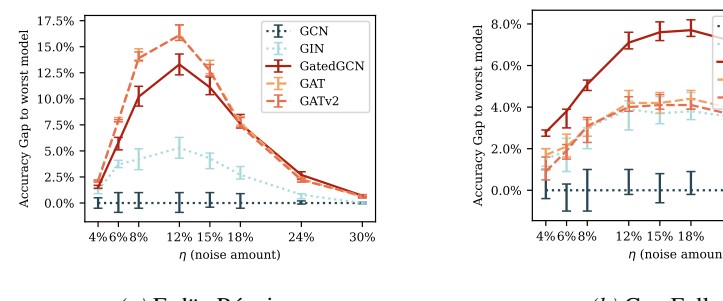
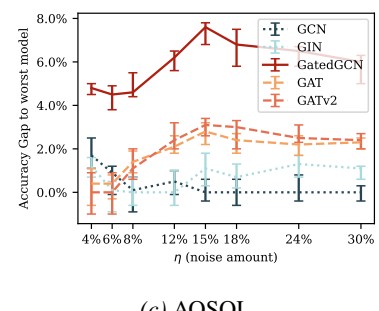

*(a)* Erdös-Rényi      *(b)* CoraFull      *(c)* AQSOL

*Figure 3.* Relative performance of well-known isotropic (GCN, GIN) and anisotropic (GatedGCN, GAT, GATv2) architectures for different graph alignment datasets based on 3 base datasets: Erdös-Rényi, CoraFull, and AQSOL.

These results demonstrate the effectiveness of the proposed graph alignment task for benchmarking architectures as well as the robustness of the results even if the models are only compared at the optimal noise level. When benchmarking a new architecture, we must choose which datasets to use. For general-purpose GNNs, the Erdös-Rényi graph alignment dataset provides a solid assessment of their structural analysis capabilities. Conversely, for application-specific architectures, we can select or generate a graph alignment dataset with graph topologies that closely match the expected characteristics of our target use case. For example, if we are working on a new architecture specialized for molecular chemistry, we will choose to benchmark our model against the graph alignment dataset based on AQSOL or PCQM4Mv2 at the optimal noise level $\eta = 15\%$. Based on our results, we are confident that across all noise levels, our model will be ranked consistently and that the task's difficulty will be optimal for mitigating variance in training and diminishing the sensitivity of the model to hyperparameter choices. While $\eta^*$ admits no closed form, it varies little within a topological family, prior values transfer as warm starts, removing the need for a full sweep.

Our results provide a more nuanced perspective on the comparison between isotropic and anisotropic GNNs (Dwivedi & Bresson, 2020). Specifically, we show that within the class of 1-WL GNNs, anisotropic architectures (such as GatedGCN, GAT, and GATv2) enable a finer characterization of structural features than isotropic ones (such as GCN and GIN)[1]. Prior work did not clearly establish whether the empirical advantages of anisotropic GNNs stemmed solely from their more precise use of node and edge features, or from anisotropy itself as a property that enhances the representation of graph topology. Moreover, our findings indicate that even on homophilic graphs, anisotropic architectures exploit richer structural representations.

**Robustness and Generalization.** Benchmarks provide insightful feedback on which architectures work best for a given task. However, it is not clear that a model that performs well on task A will also perform well on task B, even if the underlying graph topologies of both tasks are similar. In our benchmarks, we are only evaluating the structural analysis capabilities of GNNs, so that the results will generalize well to different tasks that require the same structural analysis capabilities - Figure 1. This is particularly important because, while the number and variety of tasks we are tackling in machine learning grows rapidly, these new tasks often share the same graph topologies (see Section O).

## 6. GAPE: A rich representation of the graph structure

A major risk with synthetic benchmarks (e.g., purely combinatorial tasks) is that they may encourage models to overfit to arbitrary rules that have no relevance to real-world graph properties. In this section, we show that the Graph Alignment task yields representations that generalize to diverse downstream tasks. This shows that our benchmark is measuring fundamental structural capabilities rather than just "puzzle-solving" a specific task.

### 6.1. Generating GAPE

The Siamese architecture shown in Figure 2 learns node embeddings $\boldsymbol{X} \in \mathbb{R}^{N \times d}$ as an intermediate step for a graph $\mathcal{G}$, assigning each node $i$ a vector of dimension $d$ that can serve as a positional embedding. To obtain high-quality positional embeddings, however, it is crucial to carefully select both the GNN architecture used for the Graph Alignment task and the corresponding dataset, including the base graph dataset and noise level.

In the following experiments, we define the best-performing architecture as the one achieving the highest alignment accuracy, indicating a better representation of the graph structure. All models are trained at the optimal noise level specified in Section 5, which intuitively balances accurate graph representation with robustness. Figure K.2 confirms this, showing

---

[1]Definitions are available in Section F.

*Table 1.* Comparison of GAPE against widely used positional encodings for three different molecular regression tasks using a transformer architecture.

|  | AQSOL (MAE) | PCQM4Mv2 (MAE) | ZINC (MAE) |
|---|---|---|---|
| **None** | $1.71 \pm 0.05$ | $0.236 \pm 0.004$ | $0.658 \pm 0.009$ |
| **LAP** | $1.31 \pm 0.006$ | $0.155 \pm 0.003$ | $0.171 \pm 0.008$ |
| **ABS-LAP** | $1.27 \pm 0.009$ | $0.162 \pm 0.004$ | $0.208 \pm 0.012$ |
| **R-LAP** | $1.23 \pm 0.007$ | $0.191 \pm 0.008$ | $0.50 \pm 0.042$ |
| **RWPE** | $1.12 \pm 0.004$ | $0.158 \pm 0.005$ | $0.169 \pm 0.003$ |
| **SignNet** | $1.26 \pm 0.006$ | $0.137 \pm 0.003$ | $0.116 \pm 0.005$ |
| **GPSE** | $1.105 \pm 0.008$ | $0.135 \pm 0.005$ | $\mathbf{0.104 \pm 0.006}$ |
| **GAPE** | $1.085 \pm 0.009$ | $0.133 \pm 0.003$ | $0.147 \pm 0.008$ |
| **GAPE + RWPE** | $\mathbf{1.069 \pm 0.009}$ | $\mathbf{0.125 \pm 0.004}$ | $0.109 \pm 0.005$ |

that models trained at the optimal noise level generalize well across all other noise levels.

The choice of base dataset also significantly affects the quality of the positional encodings. To illustrate this, we evaluate performance on two molecular regression tasks using the PCQM4Mv2 and AQSOL datasets. In Table K.11, we compare GAPE models pre-trained on different Graph Alignment datasets. GAPE performs best when pre-trained on the same dataset as the regression task, while mismatched topologies (e.g., OGBN-Arxiv) degrade performance, underscoring the importance of dataset choice.

### 6.2. GAPE vs other positional encodings

Graph Neural Networks (GNNs) are the standard architecture for graph-structured data. However, following the success of transformers in other domains (Vaswani et al., 2017; Dosovitskiy et al., 2021), they have been adapted to graph classification and regression tasks (Dwivedi & Bresson, 2020), particularly in molecular chemistry (Łukasz Maziarka et al., 2020; Hussain et al., 2021; Ying et al., 2021a; Rampášek et al., 2022). Applying transformers to graphs requires encoding structural information through positional encodings, which remains a challenging problem. Various methods have been proposed, and in Section 6.2, we compare Graph Alignment Positional Encodings (GAPE) to commonly used alternatives on molecular regression tasks. We justify our dataset choices and architecture choices in Section J.

Given the critical role of positional encodings in Graph Transformers, numerous approaches have been proposed in recent years. Spectral methods, such as Laplacian eigenvectors (Dwivedi & Bresson, 2020), are widely used but suffer from non-uniqueness issues. Several variants address this

limitation, including Absolute Laplacian PE (ABS-LAP), Randomized Laplacian PE (R-LAP), SignNet (Lim et al., 2022), Stable and Expressive Positional Encodings (SPE) (Huang et al., 2025), and PEARL (Kanatsoulis et al., 2025). Beyond spectral methods, Random Walk Positional Encodings (RWPE) (Ma et al., 2023) and and GPSE (Cantürk et al., 2024) are also commonly used, particularly for molecular regression tasks.

**Experimental results** To showcase the performances of GAPE, we compare it to other types of positional encodings. Following (Vaswani et al., 2017), we use a transformer architecture where each molecular graph is represented as a sequence of atoms. Positional embeddings are added to the atom embeddings to incorporate the graph structural information, and the task is to predict a given chemical property for each molecule. A detailed description of each dataset and task is provided in Section L.2.1, and the corresponding hyperparameters are listed in Section L.2.2. We use only the 2D structure of molecular graphs without edge features to ensure that the transformer relies solely on the positional encodings for structural information. For each dataset, the model is trained until convergence, and the experiment is repeated four times. In Table 1, we report the average mean absolute error (MAE) on the validation set, along with the standard deviation. In addition to standard positional encodings, we evaluate GAPE and a variant that incorporates RWPE, which offers greater expressive power than the 1-WL test. We also implemented SPE (Huang et al., 2025) and PEARL (Kanatsoulis et al., 2025), but both methods resulted in out-of-memory (OOM) errors on the PCQM4Mv2 and ZINC datasets, and are therefore omitted[2] from Table 1.

**Analysis** These results underscore the effectiveness of our approach, with GAPE+RWPE achieving the best performance on all datasets except ZINC. While GPSE is performing better than our approach on ZINC, it benefits from an unfair advantage on ZINC, as its pre-training objectives include predicting the number of cycles, which is an explicit component of the ZINC target equation. Additionally, using GAPE alone outperformed all other positional encodings on both the AQSOL and PCQM4Mv2 datasets. It is important to note that the Graph Alignment positional encodings used in these experiments is limited by the theoretical expressivity of the 1-WL[3] test, while all other evaluated positional encodings are strictly more expressive, yet GAPE still achieves competitive or better results.

---

[2]On the AQSOL dataset, both methods ran successfully but performed worse than SignNet.

[3]The GatedGCN architecture is a MPNN limited by the 1-WL, however our framework is flexible enough to work with theoretically more expressive GNNs

*Table 2.* Accuracy and F1 score on edge reconstruction of an autoencoder architecture with a frozen encoder that was trained on the graph alignment task.

| Dataset | Pre-training Dataset | Reconstruction Accuracy | Reconstruction F1 Score |
|---------|---------------------|------------------------|-------------------------|
| **ZINC** | ZINC | 98.8% | 94.7% |
| | CoraFull | 98.7% | 94.0% |
| | PCQM4Mv2 | 98.6% | 94.6% |
| | OGBNArxiv | 98.7% | 94.4% |
| **PCQM4Mv2** | ZINC | 97.7% | 94.5% |
| | CoraFull | 97.7% | 94.0% |
| | PCQM4Mv2 | 97.8% | 94.5% |
| | OGBNArxiv | 97.8% | 94.3% |

### 6.3. Graph Alignment pre-training compresses the adjacency matrix

In Section 6.2, we showed that GAPE encodes rich structural representations that improve the performance of transformers on downstream regression tasks. To further demonstrate that, although our task is rooted in a combinatorial optimization problem, it nonetheless provides a meaningful testbed for evaluating the representational capacity of GNNs across a broad range of tasks, we assess the information content of GAPE embeddings by attempting to reconstruct the original graph structure using only their latent representations.

Generative methods are widely used in molecular chemistry for applications such as drug discovery and materials design. We conduct our experiments on ZINC and PCQM4Mv2, which together cover a diverse range of molecular structures, with ZINC specifically containing synthesizable compounds curated for practical applicability.

**Experimental Setup**  Our evaluation employs an autoencoder framework comprising a frozen encoder and a learnable decoder. The encoder is a GatedGCN architecture pre-trained on the Graph Alignment task at the optimal noise level, transforming the input graph into a node feature matrix $X$ (see Figure 2). Because the encoder is frozen, any structural information available to the decoder must originate from GAPE. The decoder, a standard Transformer, is trained via Cross-Entropy loss to reconstruct the graph's adjacency matrix. Table 2 reports the edge reconstruction Accuracy and F1 score. In Figure M.3, we provide a comparison with a random baseline.

**Analysis**  We observe near-perfect reconstruction of the adjacency matrix using only GAPE. This indicates that GAPE effectively captures the complete graph topology, achieving an F1 score of $> 94\%$ on edge reconstruction. Notably, this high performance persists even when GAPE is pre-trained on different Graph Alignment datasets, highlighting the robustness and transferability of the learned structural representations.

This confirms that Graph Alignment is not just an arbitrary puzzle, but a fundamental pre-training objective that forces the model to learn a high-fidelity compression of the graph structure.

## 7. Limitations

While our approach yields promising results, several limitations remain for future exploration. First, can GNNs match the performance of combinatorial optimization algorithms on the Graph Alignment Problem (particularly for non-correlated graph alignment datasets)? Second, since our work is limited to message-passing neural networks (MPNNs), which are constrained by the expressive power of the 1-WL test, could leveraging more expressive architectures enhance both alignment accuracy and the quality of the resulting positional encodings? Note that the positional encodings against which we compare GAPE in Section 6.2 already exceed 1-WL in power.

## 8. Conclusion

We propose a new benchmarking methodology for Graph Neural Networks (GNNs) based on a *Graph Alignment Task*, emphasizing structural understanding over feature-centric approaches. Contrary to existing combinatorial optimization datasets, which are not versatile enough to encompass a broad spectrum of graph topologies and lack complexity for thorough benchmarking, our approach offers flexible difficulty levels and enables the benchmarking of GNNs on graphs from various domains, and establishes a robust framework for precise GNN evaluation. Our results indicate that each task has an optimal difficulty level where the statistical confidence in the model's performance is maximized. At this level, the best-performing architecture remains consistent across all difficulty levels.

We also show that node embeddings learned through the

*Graph Alignment Task* can serve as effective positional encodings for Transformer architectures. These embeddings contain a more accurate structural representation of the graph than other positional encodings. Additionally, we show that pre-training on the Graph Alignment task enables a high-fidelity compression of the adjacency matrix into positional encodings.

## Impact Statement

This paper presents work whose goal is to advance the field of Machine Learning. There are many potential societal consequences of our work, none which we feel must be specifically highlighted here.

## Acknowledgement

This work was partially supported by the group Casino/ENS Chair on Algorithmics and Machine Learning. The authors are grateful to the CLEPS infrastructure from INRIA Paris for providing resources and support.

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

## A. Desired Properties of the Siamese Algorithm for Graph Alignment

In Section 3 we present the siamese algorithm for training GNNs to solve the graph alignment task. This choice is motivated by the fact that it satisfies an essential property.

**Theorem A.1.** *Let $\mathcal{G}$ and $\tilde{\mathcal{G}}$ be two graphs with $N$ vertices such that $\pi^*$ is a solution of the Graph Alignment Problem between $\mathcal{G}$ and $\tilde{\mathcal{G}}$. For $\pi \in \mathcal{S}_N$, let $\mathcal{G}' = \pi \circ \mathcal{G}$. If $\mathrm{GNN}(\mathcal{G}) = \boldsymbol{X}$, $\mathrm{GNN}(\tilde{\mathcal{G}}) = \tilde{\boldsymbol{X}}$ and $\mathrm{GNN}(\mathcal{G}') = \boldsymbol{X}'$, we have the following properties:*

1. *The permutation $\pi^* \circ \pi$ is a solution to the Graph Alignment Problem between $\mathcal{G}'$ and $\tilde{\mathcal{G}}$. Hence the Graph Alignment Problem is equivariant.*

2. *The binary cross-entropy loss between the similarity matrix and the optimal permutation is invariant:*
$$\mathcal{L}(\boldsymbol{X}\tilde{\boldsymbol{X}}^T, \pi^*) = \mathcal{L}(\boldsymbol{X}'\tilde{\boldsymbol{X}}^T, \pi^* \circ \pi)$$

3. *The Linear Assignment Problem on the similarity matrix is equivariant:*
$$\mathrm{LAP}(\boldsymbol{X}'\tilde{\boldsymbol{X}}^T) = \mathrm{LAP}(\boldsymbol{X}\tilde{\boldsymbol{X}}^T) \circ \pi$$

Theorem A.1 assures us that the siamese algorithm is consistent with the equivariance property of the *Graph Alignment Problem*. Indeed, we don't want our loss $\mathcal{L}$ to be dependent on the particular labeling choice of our graph.

*Proof.* We use the same notations as in Theorem A.1 and prove each properties:

1. Let $\boldsymbol{A}$, $\tilde{\boldsymbol{A}}$ and $\boldsymbol{A}'$ be the adjacency matrices of $\mathcal{G}$, $\tilde{\mathcal{G}}$ and $\mathcal{G}'$, let $\boldsymbol{P}^*$ and $\boldsymbol{P}$ be the matrix representation of $\pi^*$ and $\pi$. By definition, because $\mathcal{G}' = \pi \circ \mathcal{G}$ then $\boldsymbol{A}' = \boldsymbol{P}\boldsymbol{A}\boldsymbol{P}^T$. We know that $\pi^*$ is a solution to the *Graph Alignment Problem* between $\mathcal{G}$ and $\tilde{\mathcal{G}}$, therefore:

$$\sum_{i,j \in \{1,\ldots,N\}} \boldsymbol{A}_{i,j}\tilde{\boldsymbol{A}}_{\pi^*(i),\pi^*(j)} = \sum_{i,j \in \{1,\ldots,N\}} (\boldsymbol{P}^T\boldsymbol{A}'\boldsymbol{P})_{i,j}\tilde{\boldsymbol{A}}_{\pi^*(i),\pi^*(j)}$$
$$= \sum_{i,j \in \{1,\ldots,N\}} \boldsymbol{A}'_{\pi^{-1}(i),\pi^{-1}(j)}\tilde{\boldsymbol{A}}_{\pi^*(i),\pi^*(j)}$$
$$= \sum_{i,j \in \{1,\ldots,N\}} \boldsymbol{A}'_{i,j}\tilde{\boldsymbol{A}}_{\pi^* \circ \pi(i),\pi^* \circ \pi(j)}$$

   Hence $\pi^* \circ \pi$ solves the *Graph Alignment Problem* between $\mathcal{G}'$ and $\tilde{\mathcal{G}}$.

2. Because GNN is by definition equivariant, we know that $\boldsymbol{X}' = \boldsymbol{P}\boldsymbol{X}$, therefore:
$$\mathcal{L}(\boldsymbol{X}'\tilde{\boldsymbol{X}}^T, \pi^* \circ \pi) = \mathcal{L}(\boldsymbol{P}\boldsymbol{X}\tilde{\boldsymbol{X}}^T, \pi^* \circ \pi) = \mathcal{L}(\boldsymbol{X}\tilde{\boldsymbol{X}}^T, \pi^* \circ \pi \circ \pi^{-1})$$
$$= \mathcal{L}(\boldsymbol{X}\tilde{\boldsymbol{X}}^T, \pi^*)$$

   Hence, the binary cross-entropy loss is invariant.

3. By using again the equivariance of GNN which states that $\boldsymbol{X}' = \boldsymbol{P}\boldsymbol{X}$, we deduce:

$$\mathrm{LAP}(\boldsymbol{X}'\tilde{\boldsymbol{X}}^T) = \left\{ \pi^* \mid \pi^* \in \arg\max_{\pi_v \in \mathcal{S}_N} \sum_{i \in \{1,\ldots,N\}} (\boldsymbol{X}'\tilde{\boldsymbol{X}}^T)_{i,\pi_v(i)} \right\}$$
$$= \left\{ \pi^* \mid \pi^* \in \arg\max_{\pi_v \in \mathcal{S}_N} \sum_{i \in \{1,\ldots,N\}} (\boldsymbol{P}\boldsymbol{X}\tilde{\boldsymbol{X}}^T)_{i,\pi_v(i)} \right\}$$
$$= \left\{ \pi^* \mid \pi^* \in \arg\max_{\pi_v \in \mathcal{S}_N} \sum_{i \in \{1,\ldots,N\}} (\boldsymbol{X}\tilde{\boldsymbol{X}}^T)_{i,\pi_v \circ \pi^{-1}(i)} \right\}$$
$$= \mathrm{LAP}(\boldsymbol{X}\tilde{\boldsymbol{X}}^T) \circ \pi$$

$\square$

## B. Graph automorphisms and the Graph Alignment task

As we train the Siamese architecture in a self-supervised manner (i.e., using a single optimal permutation $\pi$), the loss function does not explicitly account for multiple optimal permutations that arise when the automorphism group of $\mathcal{G}$ or $\tilde{\mathcal{G}}$ is non-trivial. However, in Theorem B.2 we prove that thanks to the equivariance property of the siamese architecture described in Theorem A.1, the global framework does work with graphs having non-trivial automorphism groups.

*Remark* B.1. If $\pi$ is an optimal permutation for aligning $\mathcal{G}$ and $\tilde{\mathcal{G}}$ then for all automorphism $\rho \in \text{Aut}(\mathcal{G})$, $\pi \circ \rho$ is also an optimal permutation.

**Theorem B.2.** *Let $\mathcal{G}$ be a graph with its associated automorphism group $\text{Aut}(\mathcal{G})$ and $\tilde{\mathcal{G}}$ another graph. Following the notations in Figure 2, we denote by $\boldsymbol{X}$ and $\tilde{\boldsymbol{X}}$ the node embeddings obtained after the forward pass in the GNN architecture and $\Sigma = \boldsymbol{X}\tilde{\boldsymbol{X}}^T$.*

*For any permutation $\pi$ and automorphism $\rho \in \text{Aut}(\mathcal{G})$ we have:*

$$\mathcal{L}(\Sigma, \pi) = \mathcal{L}(\Sigma, \pi \circ \rho)$$

*Proof.* We denote by $\boldsymbol{R}$ the permutation matrix associated with $\rho$.

$$
\begin{aligned}
\mathcal{L}(\Sigma, \pi \circ \rho) &= -\sum_{i=1}^{n} \ln\left(\sigma(\Sigma)\right)_{i, \pi \circ \rho(i)} \\
&= -\sum_{i=1}^{n} \ln\left(\sigma(\Sigma)\right)_{\rho^{-1}(i), \pi(i)} \\
&= -\sum_{i=1}^{n} \ln\left(\sigma(\boldsymbol{R}^T \Sigma)\right)_{i, \pi(i)} \\
&= \mathcal{L}(\boldsymbol{R}^T \Sigma, \pi)
\end{aligned}
$$

It suffices now to prove that $\boldsymbol{R}^T \Sigma = \Sigma$.

First, we notice that $\boldsymbol{R}^T \Sigma = (\boldsymbol{R}^T \boldsymbol{X})\tilde{\boldsymbol{X}}$, therefore, due to the equivariance of the GNN architecture and that $\rho^{-1} \in \text{Aut}(\mathcal{G})$, we have:

$$
\begin{aligned}
(\boldsymbol{R}^T \boldsymbol{X}) &= \boldsymbol{R}^T \text{GNN}(\mathcal{G}) \\
&= \text{GNN}(\rho^{-1} \circ \mathcal{G}) \\
&= \text{GNN}(\mathcal{G}) \\
&= \boldsymbol{X}
\end{aligned}
$$

Hence, we proved that $\boldsymbol{R}^T \Sigma = \Sigma$ and $\mathcal{L}(\Sigma, \pi) = \mathcal{L}(\Sigma, \pi \circ \rho)$. $\qquad\square$

# C. Analysis of the Graph Alignment task

## C.1. Theoretical recovery of the optimal alignment

In Section 4, we describe how to generate Graph Alignment datasets with a planted solution of the optimal permutation. However, this solution is not always guaranteed to be truly optimal: for very small graphs or when applying a noise level close to $100\%$, the current alignment may no longer remain the best one. Nevertheless, (Ganassali, 2022) provides a theoretical analysis of the optimal alignment for correlated Erdös-Renyi graphs. As an approximation, for Erdös-Renyi graphs, when the number of nodes $N$ becomes large, if

$$\eta < 1 - \frac{1}{\sqrt{Np}},$$

our framework will recover the optimal alignment. In this expression, $N$ denotes the number of nodes and $p$ the edge probability. While this result holds for Erdös-Renyi graphs, no equivalent results are known for the other graph structures considered in this paper; nonetheless, this bound can serve as a guideline. In the following table, we compute this bound using the statistics of the datasets employed in our study:

*Table C.1.* Computation of the bound for recovering the optimal alignment for each dataset used in this paper

| Base dataset | Avg order | Avg degree | Bound |
|---|---|---|---|
| Erdös-Rényi | 100 | 8.00 | 65% |
| AQSOL | 15.5 | 2.01 | 30% |
| PCQM4Mv2 | 14 | 2.02 | 30% |
| CoraFull | 100 | 4.26 | 51% |
| OGBN-Arxiv | 100 | 3.60 | 48% |

As shown in C.1, if we keep $\eta < 30\%$, then for all datasets we should, with high probability, generate instances where the planted alignment is indeed the optimal one.

## C.2. What Sets the Graph Alignment Benchmark Apart

### C.2.1. PROPERTIES OF THE GRAPH ALIGNMENT TASK

In this paper, we emphasize the importance of ensuring that the following properties are present in our benchmark:

- Isolating and benchmarking solely the structural understanding of GNNs.

- Applicability to any existing graph dataset without modifying the topology, thereby allowing us to reveal how different graph structures influence model performance.

- Variable difficulty levels, which we demonstrate are essential for statistically meaningful comparisons (see Section 5).

Incorporating all these features into a benchmark imposes strong restrictions on the set of tasks we can consider. Combinatorial Optimization tasks on graphs naturally arise as a candidate, since they allow us to benchmark only the structural understanding of a model. However, all combinatorial tasks previously used with GNNs fail to satisfy the remaining properties:

*Table C.2.* Limitations of current Combinatorial Optimization based benchmarks

| Dataset | Description | Limitations |
| --- | --- | --- |
| **G4satbench** Li et al. (2024) | Benchmark based on the SAT problem | Inability to adapt to a wide variety of graph topologies |
| **PATTERN** Dwivedi et al. (2022) | Finding a fixed graph pattern P embedded in larger graphs | Only based on SBM graph topologies |
| **CLUSTER** Dwivedi et al. (2022) | Clustering SBM graphs generated with 6 communities | Only based on SBM graph topologies |
| **CSL** Dwivedi et al. (2022) | Classification on regular undirected graphs with 41 nodes where each node has degree 4 | Synthetic regular graphs topologies are not representative of all graph topologies |
| **TSP** Dwivedi et al. (2022) | Solving the TSP problem on random instances with GNNs | Synthetic datasets as well as inability to control problem difficulty. |
| Chen et al. (2020) | Counting substructures in graphs | Inability to control the task difficulty |
| Zhang et al. (2024) | Tasks related to 2-connectedness in graphs | Inability to control the task difficulty |
| Li et al. (2022) | Find the chromatic number of a graph and the coloring. | Inability to control the task difficulty |

Compared to the combinatorial tasks previously used to benchmark GNNs, key design choices allow the Graph Alignment task to satisfy all the aforementioned properties:

- The Siamese architecture employed in this paper to solve the Graph Alignment task relies exclusively on the connectivity patterns of the graphs.

- In Section 4, we describe how to generate Graph Alignment datasets from any base dataset (ranging from a single large graph dataset to molecular datasets containing millions of graphs). Moreover, the Graph Alignment task is a combinatorial problem defined on a pair of graphs (whereas other combinatorial tasks used to benchmark GNNs are typically defined on a single graph). This setting enables us to plant a solution (required for NP-hard problems) without altering the topology of the graph.

- The noising procedure introduced in Section 4 provides a direct mechanism to control the difficulty of the task.

C.2.2. A METRIC LEARNING INTERPRETATION

In addition, the Graph Alignment task can also be justified by viewing it as a metric learning problem. Consider a distance $d$ on the space of unlabeled graphs. Then, there exists a distance $\tilde{d}$ on the space of labeled graphs such that, for $\mathcal{G}$ and $\tilde{\mathcal{G}}$ with

adjacency matrices $\boldsymbol{A}$ and $\tilde{\boldsymbol{A}}$:

$$d(\mathcal{G}, \tilde{\mathcal{G}}) = \min_{P \in \mathcal{S}} \tilde{d}(\boldsymbol{A}, \boldsymbol{P}\tilde{\boldsymbol{A}}\boldsymbol{P}^T).$$

From this formulation, we can infer that if $\mathcal{G} \cong \tilde{\mathcal{G}}$, then $\boldsymbol{P}_{\text{opt}}$, the permutation minimizing $\tilde{d}(\boldsymbol{A}, \boldsymbol{P}_{\text{opt}}\tilde{\boldsymbol{A}}\boldsymbol{P}_{\text{opt}}^T)$, is the same for any distance $d$ (and hence any $\tilde{d}$). This shows that for isomorphic graphs, computing the optimal permutation already contains all the necessary information.

More generally, if $\mathcal{G}$ and $\tilde{\mathcal{G}}$ are arbitrary graphs and we take $\tilde{d}(\boldsymbol{A}, \tilde{\boldsymbol{A}}) = \|\boldsymbol{A} - \tilde{\boldsymbol{A}}\|$, then:

$$d(\mathcal{G}, \tilde{\mathcal{G}}) = \min_{P \in \mathcal{S}} \|\boldsymbol{A} - \boldsymbol{P}\tilde{\boldsymbol{A}}\boldsymbol{P}^T\|.$$

Solving this minimization problem is equivalent to solving

$$\max_{P \in \mathcal{S}} \#\text{CommonEdges}(\boldsymbol{A}, \boldsymbol{P}\tilde{\boldsymbol{A}}\boldsymbol{P}^T) = \max_{P \in \mathcal{S}} \sum_{i,j \in \{1,\dots,N\}} \boldsymbol{A}_{i,j} \tilde{\boldsymbol{A}}_{\boldsymbol{P}(i), \boldsymbol{P}(j)},$$

which corresponds exactly to the Graph Alignment task.

## C.3. Links between the Graph Alignment task and GNN expressivity

By design, the Graph Alignment task is challenging. The metric learning interpretation of graph alignment guarantees that

$$d(\mathcal{G}, \tilde{\mathcal{G}}) = 0 \quad \Leftrightarrow \quad \mathcal{G} \cong \tilde{\mathcal{G}}.$$

As discussed, for isomorphic graphs, all the information is contained in the optimal permutation. Thus, solving the Graph Alignment task on isomorphic graphs requires solving the graph isomorphism problem, which is not known to be solvable in polynomial time. Consequently, even at $\eta = 0$ in our benchmark, GNNs with different levels of expressivity can be distinguished. This inherent difficulty is precisely what makes the task a meaningful proxy for evaluating structural understanding: at zero noise, GNNs with different theoretical expressivity (e.g., as classified by the WL test) can be differentiated, while at higher noise levels, the task highlights more subtle differences among GNNs belonging to the same expressivity class.

The following theorem also shows that at any noise, there is a link between the Graph Alignment task and GNN expressivity:

**Theorem C.1** (WL Hierarchy and Graph Alignment). *The WL hierarchy induces a strict hierarchy of achievable performance (accuracy and loss) on the Graph Alignment Problem. A GNN that assigns identical embeddings to two distinct nodes in $G$ cannot reliably align those nodes to their correct counterparts in any noisy copy $G'$, regardless of the noise level $\sigma$. The alignment error is bounded below by the number of nodes the GNN fails to distinguish.*

*Proof.* Let $f$ be a GNN that produces node embeddings $f(G, v) \in \mathbb{R}^d$ for each node $v$. Consider the equivalence classes induced by $f$ on $G$:

$$u \sim v \iff f(G, u) = f(G, v).$$

Following the notation we note $\Sigma = f(G)f(\tilde{G})^T$. Let $C_1, \dots, C_k$ be all the equivalence classes with 2 or more elements. Then for $u, v \in C_i$, $\Sigma_{u,:} = \Sigma_{v,:}$ (they are mapped to the same probability distribution over all nodes of $\tilde{G}$), hence the accuracy of mapping any $u \in C_i$ to its correct target is bounded by $1/|C_i|$.

Let $S = \sum_{i=1}^{k} |C_i|$ be the total number of nodes in non-trivial classes, and let $n$ be the total number of nodes. The remaining $n - S$ nodes are singletons under $f$ (uniquely identified). Therefore the global accuracy satisfies:

$$\text{acc}(f) \leq \frac{1}{n}\left((n-S)\cdot 1 + \sum_{i=1}^{k} |C_i| \cdot \frac{1}{|C_i|}\right) = \frac{n - S + k}{n} = 1 - \frac{S - k}{n}.$$

Similarly, we obtain:

$$L_{CE}(f) \geq \frac{1}{n}\sum_{i=1}^{k} |C_i| \log |C_i|.$$

Both bounds are tight: they can be achieved by a GNN that perfectly distinguishes all nodes outside the non-trivial classes. $\square$

**Corollary C.2** (WL hierarchy). *If $g$ is a strictly more expressive GNN that splits some $C_i$ into smaller subclasses, $k$ strictly increases and $S$ decreases, so the accuracy upper bound $\frac{n-S+k}{n}$ strictly increases and the cross-entropy lower bound strictly decreases. Each level of the WL hierarchy therefore yields a strictly better alignment bound.*

## C.4. The Alignment Task as a Denoising Problem

*Table C.3.* We experimentally verify that increasing the size of the $n$-hop neighborhood improves the signal-to-noise ratio and, consequently, the accuracy. The results are obtained on a Graph Alignment dataset based on Erdös-Renyi graphs, corrupted at the optimal noise level ($\eta = 12\%$). We report the average accuracy along with the standard deviation for a GCN architecture with varying numbers of layers (corresponding to different $n$-hop neighborhood sizes). Each result is computed over 4 runs with different random seeds.

| Layers | 2 | 3 | 4 | 5 | 6 | 7 |
|---|---|---|---|---|---|---|
| **Accuracy** | $59.6\% \pm 0.3\%$ | $64.9\% \pm 0.1\%$ | $72.6\% \pm 0.1\%$ | $77.4\% \pm 0.1\%$ | $79.0\% \pm 0.1\%$ | $79.5\% \pm 0.1\%$ |

Another possible interpretation of what the Siamese architecture learns arises when the GNN architecture is a Message-Passing Neural Network (MPNN). An MPNN updates the representation of a node $i$ by aggregating the information from its neighborhood (i.e., the representations of the neighbors of $i$) at each layer. Consequently, an MPNN with $n$ layers computes the representation of $i$ based on its $n$-hop neighborhood. From the perspective of node $i$, the noising procedure perturbs its $n$-hop neighborhood. However, in order to maximize the similarity between node $i$ in the base graph and node $i$ in the corrupted graph, the Siamese network must learn to denoise the corrupted $n$-hop neighborhood so that its representation is as close as possible to that of the base $n$-hop neighborhood. In addition, because the objective is to recover the alignment, the model is inherently prevented from representation collapse: the Siamese architecture must also learn node representations that are as discriminative as possible in order to distinguish between different nodes. This mechanism is reminiscent of student-teacher approaches such as Bootstrap Your Own Latent (Grill et al., 2020), where collapse is avoided not through negative samples but by enforcing consistency under transformations. Here, the alignment objective plays an analogous role.

Hence, we expect that, at a fixed noise level, increasing the number of layers in the MPNN architecture will enlarge the neighborhood used to compute each node's representation, thereby improving the signal-to-noise ratio. In Table C.3 we show that this indeed holds experimentally.

## D. Breadth-First Sampling

As noted in Section 4, large graphs require a subsampling step before generating the final Graph Alignment dataset. Algorithm 1 provides a method to subsample graphs while preserving their local topology.

---

**Algorithm 1** BFS Subgraph Sampling

---

**Require:** $\mathcal{G}^{big} = (V, E)$ and $N'$
**Ensure:** $\mathcal{G}'$ (subgraph of $\mathcal{G}^{big}$) with $N'$ vertices
  Initialize vertices set: $V' \leftarrow \emptyset$
  Randomly sample a vertex $i \in V$ and add $i$ to $V'$
  **while** $|V'| < N'$ **do**
    Define $A \leftarrow \{j | j \sim i \text{ for } i \in V'\}$
    **if** $|V' \cup A| > N'$ **then**
      Randomly remove $|V' \cup A| - N'$ nodes from $A$
    **end if**
    $V' \leftarrow V' \cup A$
  **end while**
  Define $\mathcal{G}' \leftarrow (V', E[V'])$, where $E[V']$ are edges in $\mathcal{G}^{big}$ induced by $V'$
  **Return** $\mathcal{G}'$

---

## E. Experimental Details: Benchmarking

In Section 5, we benchmark several models using our methodology based on the *Graph Alignment Task*. We build forty different datasets - for each of the five base datasets, we generate eight graph alignment datasets with $\eta \in \{4\%, 6\%, 8\%, 12\%, 15\%, 18\%, 24\%, 30\%\}$. In Table E.4, we summarize for each base dataset the parameters that are used to generate the associated graph alignment datasets.

*Table E.4.* Summary of the different graph alignment datasets created using our methodology. **Size** corresponds to the size of the graph alignment datasets generated and not the size of the base dataset.

| Base dataset | Size (train/val) | Avg order | Avg degree | BFS Sampling | Noising Model |
|---|---|---|---|---|---|
| Erdös-Rényi | 5,000/500 | 100 | 8.00 | No | Add + Remove |
| AQSOL | 7,836/998 | 15.5 | 2.01 | No | Add |
| PCQM4Mv2 | 35,000/2,000 | 14 | 2.02 | No | Add |
| CoraFull | 5,000/500 | 100 | 4.26 | Yes | Add + Remove |
| OGBN-Arxiv | 5,000/500 | 100 | 3.60 | Yes | Add + Remove |

In addition, we also include the key hyperparameters of the benchmarked GNNs in Table E.5.

Compared to the *vanilla* version of each of those GNNs, the GNNs that we train include residual connections and GraphNorm layers (the implementation of each model is available in the accompanying Python package). The training was performed for 300 epochs on an *NVIDIA RTX6000 24GB* GPU. We used the AdamW optimization algorithm with a One-Cycle learning rate schedule, incorporating 30 warmup steps and a maximum learning rate of 0.003. A gradient clipping value of 0.1 was applied to stabilize training. Training times ranged from 20 to 40 minutes per run, except for the larger graph alignment datasets based on PCQM4Mv2, which required 2.5 hours but reached convergence within 1 hour[4].

Every result in Section 5 can be reproduced using the Python package. We provide a command-line tool to generate all the datasets used in this paper and train our pre-defined GNN architectures or custom ones on the *Graph Alignment Task*.

---

[4]Performance did not improve by more than 0.5% afterward

*Table E.5.* Key hyperparameters of the five benchmarked models. To ensure a meaningful comparison, a fixed trainable parameter budget was allocated, and the dimension of the final layer was standardized, requiring all models to learn node embeddings of the same dimension.

|  | # Parameters | Layers | Heads | Width | Final Layer |
|---|---|---|---|---|---|
| **GGN** | 58,560 | 4 | N.A | 128 | 64 |
| **GIN** | 59,026 | 4 | N.A | 93 | 64 |
| **GatedGCN** | 59,680 | 4 | N.A | 48 | 64 |
| **GAT** | 59,328 | 4 | 8 | 128 | 64 |
| **GATv2** | 62,848 | 4 | 8 | 96 | 64 |

## F. Full Benchmarking Results

We provide in Table F.6 the absolute accuracy of every benchmarked model for each graph alignment dataset (across all base datasets and all noise levels). The median validation accuracy for ten runs is reported in percent as well as the standard deviation.

**Definition F.1** (Isotropic GNNs). Isotropic GNNs are graph neural networks where messages are a function of the source node only. These models treat every edge direction equally in the node update equation, meaning the same message function is applied regardless of the target node (see Tailor et al. (2022)).

**Definition F.2** (Anisotropic GNNs). Anisotropic GNNs are graph neural networks where messages sent between nodes are a function of both the source and target node. In these models, the message-passing mechanism treats each edge direction differently, allowing for directional information flow that depends on the specific pair of nodes involved in the communication (see Tailor et al. (2022)).

Our results show that anisotropic GNNs outperform isotropic GNNs on our benchmark.

*Table F.6.* Test accuracy (in percentage) of the benchmarked GNNs on the generated graph alignment datasets described in Section 4. Red: highlights the best result and those within 1% of it. Blue: indicates results within 3% of the best performance.

| | | Noise: $\eta$ | | | | | | | |
|---|---|---|---|---|---|---|---|---|---|
| | | 4% | 6% | 8% | 12% | 15% | 18% | 24% | 30% |
| **Erdös-Rényi** | **Laplacian** | 16.7 | 12.9 | 10.3 | 7.5 | 6.6 | 5.9 | 4.8 | 4.1 |
| | **GCN** | $97.6 \pm 0.4$ | $90.5 \pm 0.9$ | $78.9 \pm 0.9$ | $49.5 \pm 0.9$ | $34.3 \pm 0.6$ | $24.2 \pm 0.6$ | $11.7 \pm 0.1$ | $7.0 \pm 0.1$ |
| | **GIN** | $98.8 \pm 0.2$ | $94.2 \pm 0.3$ | $83.1 \pm 1.5$ | $54.8 \pm 4.2$ | $38.6 \pm 0.8$ | $26.9 \pm 0.5$ | $12.5 \pm 0.2$ | $7.0 \pm 0.1$ |
| | **GatedGCN** | $99.1 \pm 0.1$ | $96.3 \pm 0.5$ | $89.1 \pm 0.8$ | $62.8 \pm 0.9$ | $45.4 \pm 0.9$ | $31.8 \pm 0.5$ | $14.4 \pm 0.3$ | $7.7 \pm 0.1$ |
| | **GAT** | $99.8 \pm 0.0$ | $98.4 \pm 0.1$ | $92.8 \pm 0.5$ | $65.6 \pm 0.8$ | $47.0 \pm 0.6$ | $31.9 \pm 0.3$ | $14.0 \pm 0.1$ | $7.6 \pm 0.1$ |
| | **GATv2** | $99.7 \pm 0.1$ | $98.4 \pm 0.1$ | $92.8 \pm 0.3$ | $65.6 \pm 0.6$ | $47.0 \pm 0.4$ | $31.6 \pm 0.4$ | $13.9 \pm 0.1$ | $7.7 \pm 0.1$ |
| **AQSOL** | **Laplacian** | 64.4 | 54.8 | 48.7 | 40.0 | 36.6 | 33.1 | 29.4 | 27.0 |
| | **GCN** | $80.1 \pm 0.5$ | $75.2 \pm 0.5$ | $70.2 \pm 0.6$ | $61.5 \pm 0.4$ | $55.7 \pm 0.4$ | $50.2 \pm 0.5$ | $41.7 \pm 0.5$ | $34.3 \pm 0.3$ |
| | **GIN** | $79.6 \pm 0.4$ | $74.4 \pm 0.7$ | $70.1 \pm 0.5$ | $61.0 \pm 0.6$ | $56.8 \pm 0.5$ | $50.9 \pm 0.4$ | $43.0 \pm 0.4$ | $35.4 \pm 0.3$ |
| | **GatedGCN** | $83.2 \pm 0.2$ | $78.8 \pm 0.4$ | $74.7 \pm 0.4$ | $67.2 \pm 0.4$ | $63.3 \pm 0.4$ | $57.0 \pm 0.9$ | $48.2 \pm 0.4$ | $40.3 \pm 0.7$ |
| | **GAT** | $78.8 \pm 0.8$ | $74.7 \pm 0.4$ | $71.5 \pm 0.4$ | $63.1 \pm 0.3$ | $58.5 \pm 0.4$ | $52.6 \pm 0.3$ | $43.9 \pm 0.3$ | $36.6 \pm 0.2$ |
| | **GATv2** | $78.4 \pm 0.8$ | $74.3 \pm 0.8$ | $71.2 \pm 0.6$ | $63.4 \pm 0.5$ | $58.8 \pm 0.3$ | $53.2 \pm 0.5$ | $44.2 \pm 0.4$ | $36.7 \pm 0.3$ |
| **PCQM4Mv2** | **Laplacian** | 75.6 | 68.3 | 60.1 | 50.4 | 45.9 | 43.3 | 38.8 | 33.9 |
| | **GCN** | $89.1 \pm 1.8$ | $84.7 \pm 0.5$ | $81.0 \pm 0.7$ | $71.5 \pm 0.4$ | $61.7 \pm 1.7$ | $62.0 \pm 0.4$ | $52.1 \pm 0.5$ | $44.4 \pm 0.3$ |
| | **GIN** | $89.4 \pm 0.3$ | $84.9 \pm 0.4$ | $81.8 \pm 0.5$ | $72.3 \pm 0.6$ | $64.3 \pm 0.9$ | $63.6 \pm 0.5$ | $54.3 \pm 0.4$ | $46.8 \pm 0.4$ |
| | **GatedGCN** | $91.7 \pm 0.3$ | $88.2 \pm 0.3$ | $85.9 \pm 0.4$ | $78.2 \pm 0.5$ | $70.2 \pm 0.6$ | $69.9 \pm 0.6$ | $60.4 \pm 0.6$ | $52.3 \pm 0.8$ |
| | **GAT** | $88.5 \pm 0.7$ | $84.9 \pm 0.5$ | $81.9 \pm 0.3$ | $73.5 \pm 0.4$ | $66.0 \pm 0.6$ | $64.5 \pm 0.3$ | $54.7 \pm 0.4$ | $46.8 \pm 0.2$ |
| | **GATv2** | $88.6 \pm 1.0$ | $84.3 \pm 1.0$ | $81.8 \pm 0.6$ | $73.7 \pm 0.4$ | $65.7 \pm 0.5$ | $64.7 \pm 0.5$ | $55.2 \pm 0.2$ | $47.2 \pm 0.4$ |
| **OGBN-Arxiv** | **Laplacian** | 24.1 | 21.0 | 19.3 | 16.7 | 15.7 | 14.7 | 13.2 | 11.9 |
| | **GCN** | $71.0 \pm 0.3$ | $62.9 \pm 0.4$ | $54.5 \pm 0.4$ | $40.8 \pm 0.3$ | $33.0 \pm 0.4$ | $27.1 \pm 0.4$ | $18.7 \pm 0.2$ | $13.4 \pm 0.2$ |
| | **GIN** | $72.9 \pm 0.6$ | $66.3 \pm 1.0$ | $59.8 \pm 0.3$ | $47.0 \pm 0.2$ | $39.5 \pm 0.3$ | $33.1 \pm 0.2$ | $23.3 \pm 0.2$ | $16.7 \pm 0.2$ |
| | **GatedGCN** | $75.9 \pm 0.1$ | $70.2 \pm 0.1$ | $63.3 \pm 0.2$ | $51.2 \pm 0.2$ | $43.6 \pm 0.2$ | $37.0 \pm 0.3$ | $26.5 \pm 0.2$ | $18.8 \pm 0.2$ |
| | **GAT** | $72.4 \pm 0.2$ | $65.8 \pm 0.3$ | $57.9 \pm 0.3$ | $44.8 \pm 0.2$ | $37.6 \pm 0.3$ | $31.0 \pm 0.2$ | $21.5 \pm 0.2$ | $15.6 \pm 0.1$ |
| | **GATv2** | $72.4 \pm 0.3$ | $65.8 \pm 0.4$ | $58.2 \pm 0.2$ | $45.1 \pm 0.3$ | $37.9 \pm 0.4$ | $31.5 \pm 0.3$ | $22.1 \pm 0.4$ | $16.1 \pm 0.4$ |
| **CoraFull** | **Laplacian** | 18.2 | 15.3 | 13.5 | 11.4 | 10.1 | 9.0 | 8.0 | 6.9 |
| | **GCN** | $77.9 \pm 0.5$ | $72.4 \pm 0.6$ | $65.4 \pm 0.8$ | $54.6 \pm 0.6$ | $47.5 \pm 0.5$ | $39.2 \pm 0.5$ | $28.2 \pm 0.4$ | $20.6 \pm 0.3$ |
| | **GIN** | $79.5 \pm 0.1$ | $74.3 \pm 0.4$ | $66.8 \pm 0.7$ | $58.5 \pm 0.6$ | $51.2 \pm 0.5$ | $43.0 \pm 0.5$ | $31.6 \pm 0.4$ | $22.8 \pm 0.2$ |
| | **GatedGCN** | $80.7 \pm 0.1$ | $76.2 \pm 0.4$ | $70.5 \pm 0.2$ | $61.7 \pm 0.3$ | $55.1 \pm 0.3$ | $46.9 \pm 0.3$ | $35.1 \pm 0.4$ | $25.2 \pm 0.2$ |
| | **GAT** | $79.6 \pm 0.4$ | $74.5 \pm 0.5$ | $68.4 \pm 0.3$ | $58.8 \pm 0.4$ | $51.7 \pm 0.4$ | $43.6 \pm 0.3$ | $31.9 \pm 0.4$ | $22.6 \pm 0.2$ |
| | **GATv2** | $78.8 \pm 0.5$ | $74.3 \pm 0.3$ | $68.5 \pm 0.4$ | $58.6 \pm 0.3$ | $51.6 \pm 0.3$ | $43.3 \pm 0.2$ | $31.6 \pm 0.4$ | $22.8 \pm 0.2$ |

# G. Graph Alignment benchmark on extended architectures

We provide additional benchmarking results at the optimal noise level for several GNN models and different parameter budgets. These results are consistent with the results presented in Table F.6 and confirm that anisotropic GNNs have a better understanding of the graph structure than isotropic GNNs.

*Table G.7.* Model hyperparameters used to generate the results in Section G. Small models have approximately 60k parameters. Medium models have approximatly 140k parameters.

| Model Name | Width | Layers | Heads | Final Layer |
|---|---|---|---|---|
| **GAT-small** | 128 | 4 | 8 | 64 |
| **GAT-med** | 128 | 8 | 8 | 64 |
| **GATv2-small** | 96 | 4 | 8 | 64 |
| **GATv2-med** | 96 | 8 | 8 | 64 |
| **GCN-small** | 128 | 4 | - | 64 |
| **GCN-med** | 128 | 8 | - | 64 |
| **GIN-small** | 96 | 4 | - | 64 |
| **GIN-med** | 96 | 8 | - | 64 |
| **GatedGCN-small** | 48 | 4 | - | 64 |
| **GatedGCN-med** | 48 | 8 | - | 64 |
| **GraphGPS-small** | 36 | 4 | 4 | 64 |
| **GraphGPS-med** | 36 | 8 | 4 | 64 |
| **PNA-small** | 35 | 4 | - | 64 |
| **PNA-med** | 35 | 8 | - | 64 |
| **SGC-small** | 128 | 4 | - | 64 |
| **SGC-med** | 128 | 8 | - | 64 |

*Table G.8.* Extended results of our benchmark on several graph alignment datasets. We report the mean accuracy on the validation set as well as the standard deviation. Each model was run 4 times with different random seeds.

| Model | AQSOL | CoraFull | Erdós-Rényi | OGBNArxiv | PCQM4Mv2 | ZINC |
|---|---|---|---|---|---|---|
| GAT-med | 0.5579 ± 0.0065 | 0.4874 ± 0.0009 | 0.8236 ± 0.0004 | 0.3621 ± 0.0010 | 0.6034 ± 0.0011 | 0.5551 ± 0.0008 |
| GAT-small | 0.5756 ± 0.0013 | 0.4575 ± 0.0028 | 0.7396 ± 0.0011 | 0.3217 ± 0.0029 | 0.5480 ± 0.0007 | 0.4446 ± 0.0013 |
| GatedGCN-med | 0.5953 ± 0.0039 | 0.4921 ± 0.0005 | 0.7818 ± 0.0014 | 0.3827 ± 0.0012 | 0.6248 ± 0.0005 | 0.5896 ± 0.0006 |
| GatedGCN-small | 0.6304 ± 0.0011 | 0.4921 ± 0.0007 | 0.7850 ± 0.0020 | 0.3785 ± 0.0010 | 0.6134 ± 0.0019 | 0.5504 ± 0.0004 |
| GATv2-med | 0.5614 ± 0.0079 | 0.4877 ± 0.0013 | 0.8239 ± 0.0004 | 0.3758 ± 0.0032 | 0.6081 ± 0.0014 | 0.5657 ± 0.0013 |
| GATv2-small | 0.5842 ± 0.0037 | 0.4626 ± 0.0021 | 0.7473 ± 0.0006 | 0.3305 ± 0.0017 | 0.5546 ± 0.0010 | 0.4544 ± 0.0008 |
| GCN-med | 0.5707 ± 0.0017 | 0.4823 ± 0.0009 | 0.8184 ± 0.0009 | 0.3350 ± 0.0011 | 0.5993 ± 0.0011 | 0.5373 ± 0.0017 |
| GCN-small | 0.5502 ± 0.0041 | 0.4231 ± 0.0013 | 0.6429 ± 0.0038 | 0.2810 ± 0.0007 | 0.5269 ± 0.0006 | 0.4078 ± 0.0003 |
| GIN-med | 0.5744 ± 0.0008 | 0.4784 ± 0.0013 | 0.8181 ± 0.0016 | 0.3573 ± 0.0006 | 0.6058 ± 0.0017 | 0.5665 ± 0.0012 |
| GIN-small | 0.5564 ± 0.0050 | 0.4741 ± 0.0007 | 0.7293 ± 0.0011 | 0.3426 ± 0.0007 | 0.5585 ± 0.0006 | 0.4489 ± 0.0011 |
| GraphGPS-med | 0.6070 ± 0.0046 | 0.4784 ± 0.0010 | 0.7184 ± 0.0008 | 0.3610 ± 0.0022 | 0.6051 ± 0.0013 | 0.5489 ± 0.0075 |
| GraphGPS-small | 0.6248 ± 0.0043 | 0.4741 ± 0.0018 | 0.7234 ± 0.0032 | 0.3584 ± 0.0017 | 0.5907 ± 0.0012 | 0.5181 ± 0.0039 |
| PNA-med | 0.6272 ± 0.0042 | 0.4985 ± 0.0010 | 0.7151 ± 0.0024 | 0.4030 ± 0.0005 | 0.6139 ± 0.0024 | 0.5651 ± 0.0015 |
| PNA-small | 0.6090 ± 0.0009 | 0.4791 ± 0.0012 | 0.6995 ± 0.0016 | 0.3605 ± 0.0008 | 0.5715 ± 0.0016 | 0.4768 ± 0.0004 |
| SGC-med | 0.5428 ± 0.0032 | 0.4341 ± 0.0013 | 0.6967 ± 0.0005 | 0.2777 ± 0.0019 | 0.5741 ± 0.0022 | 0.5047 ± 0.0012 |
| SGC-small | 0.5770 ± 0.0029 | 0.4034 ± 0.0013 | 0.5873 ± 0.0023 | 0.2550 ± 0.0013 | 0.5373 ± 0.0009 | 0.4303 ± 0.0005 |

## H. Graph Alignment benchmark for higher-order GNNs

*Table H.9.* Comparison of higher-order GNNs architecture on the Graph Alignment Task for five different base datasets.

| | AQSOL | CoraFull | Erdös-Rényi | OGBNArxiv | PCQM4Mv2 | ZINC |
|---|---|---|---|---|---|---|
| GatedGCN | 0.58 | **0.46** | **0.74** | 0.37 | 0.61 | 0.55 |
| GSN | 0.58 | **0.46** | 0.71 | 0.36 | 0.58 | 0.52 |
| PPGN | **0.61** | 0.44 | 0.45 | **0.77** | **0.64** | **0.64** |

We ran additional experiments higher orders GNNs: GSN (Bouritsas et al., 2020) and PPGN (Maron et al., 2019) which are both as expressive as 3-WL. PPGN substantially outperforms 1-WL models on several datasets (OGBN-Arxiv, ZINC, PCQM4Mv2, AQSOL), while performing comparably to GatedGCN on CoraFull. This is consistent with our benchmark's design: datasets where topology is more complex expose the expressivity gap between WL classes, while simpler topologies can be resolved at the 1-WL level. These results confirm that our benchmark can discriminate both within and across WL

classes.

## I. Correlation of the Graph Alignment problem with the downstream tasks

To validate that the structural understanding benchmarked via the Graph Alignment task translates to meaningful improvements in downstream applications, we analyze the correlation between a model's performance on our GA benchmark and its predictive accuracy on established molecular property prediction datasets.

We evaluate a diverse set of model across two scales ("Small" and "Medium") on both the ZINC and PCQM4Mv2 datasets. For each configuration, we record:

- **Graph Alignment Accuracy:** The model's ability to solve the Graph Alignment task (higher is better).

- **Downstream Performance:** The Mean Absolute Error (MAE) on the target regression task (lower is better).

Figure I.1 visualizes the relationship between these two metrics. We observe a strong negative Pearson correlation ($r < -0.90$) across all settings. Specifically:

- **Strong Predictive Power:** Models that achieve higher accuracy on the Graph Alignment task consistently yield lower error rates (MAE) on downstream tasks. This trend holds true for both small-scale (ZINC) and large-scale (PCQM4Mv2) benchmarks.

- **Structural Bottleneck:** The tight linearity of the data points suggests that structural reasoning capability is a primary bottleneck for these tasks. Improvements in resolving graph structural patterns directly translate to better molecular property prediction.

- **Robustness to Scale:** The correlation remains significant regardless of model size (Small vs. Medium), indicating that the Graph Alignment task is a reliable proxy for measuring the GNN ability to understand the topology of the graph, independent of parameter count.

These findings empirically confirm that Graph Alignment is not merely an auxiliary puzzle but a fundamental pre-training objective. By forcing the model to solve the alignment problem, we ensure it learns a robust, globally consistent representation of graph topology that is critical for high-performance molecular modeling.

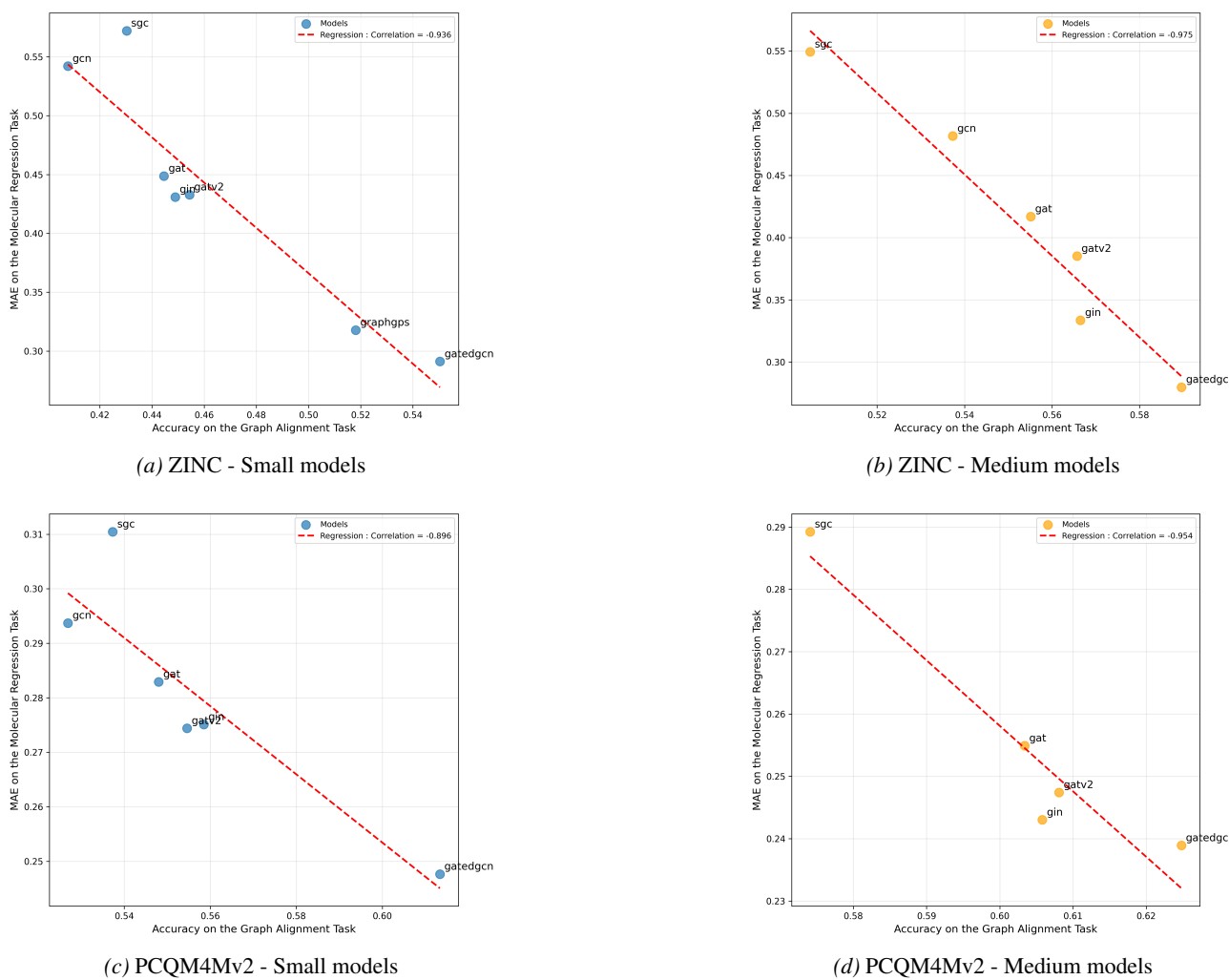

*(a)* ZINC - Small models

*(b)* ZINC - Medium models

*(c)* PCQM4Mv2 - Small models

*(d)* PCQM4Mv2 - Medium models

*Figure I.1.* Probing the Pearson Correlation between accuracy on the Graph Alignment Task and MAE on the PCQM4Mv2 and ZINC datasets. The strong negative correlation indicates that better structural alignment capability leads to lower downstream error.

## J. Architecture and Dataset Selection for Evaluating GAPE

Many benchmarks use combinatorial optimization tasks to compare different GNN architectures. However, as described in Section 3 and Section C.2.1, these benchmarks often fail to generalize to existing, real-world datasets, making it difficult to determine whether the models learn meaningful graph representations or simply specialize on the combinatorial task they were trained on. In contrast, our benchmark is adaptable to any graph dataset. As shown in Section C.2.2 and Section C.4, learning the optimal alignment between two graphs can be interpreted as either a metric learning task or a denoising task, both frameworks well-suited to representational learning. Consequently, it is natural to experimentally verify that the learned embeddings capture a rich representation of the structural properties of each node in a graph.

To verify this claim, we naturally choose the Transformer architecture. Over the past few years, this architecture has become the *de-facto* standard for many tasks in deep learning. Moreover, the Transformer is equivariant and does not have inherent access to the structure of the graph. To incorporate graph structure into the Transformer, it is necessary to use positional encodings. However, unlike text or images, no canonical positional encodings exist for graphs, and many recent works propose novel encodings to better capture graph structure. This makes the Transformer a particularly suitable testbed for quantifying the quality of the representations learned during the Graph Alignment task.

Since the complexity of computing attention is $O(n^2)$, vanilla Transformers or Transformers implementing global attention have mostly been applied to graphs with a relatively small number of nodes, particularly in molecular regression tasks.

Notably, on the PCQM4Mv2 leaderboard, Transformer-based architectures perform especially well compared to the other two OGB Large Scale Challenge datasets. Furthermore, due to the popularity of molecular regression benchmarks in the graph learning community, a wide range of positional encodings have been evaluated on them, allowing us to compare GAPE to a broader set of methods. For these reasons, we selected AQSOL, ZINC, and PCQM4Mv2 to demonstrate the quality of the positional encodings learned during the Graph Alignment task.

## K. How to choose the right pre-training dataset for GAPE?

Generating GAPE requires selecting a pre-trained model. While choosing the best-performing architecture is a natural choice, we must also consider the graph alignment dataset (*i.e.*, graph topologies and noise level) on which it was trained. We argue that selecting a Graph Alignment dataset generate at the optimal noise is the right choice, in Figure K.2, we show that if the an architecture trained at the optimal noise generalize well to all other noise levels, hence will be more robust to out-of-distribution samples (*i.e.* graphs not in the Graph Alignment dataset).

In addition, to that it is also necessary to choose the right base dataset that will be used to create the Graph Alignment dataset. We show in Table K.11, that using a base dataset with similar graph topologies yields to optimal performances.

We evaluate the performance of GAPE pre-trained on different Graph Alignment datasets against MPNNs and Laplacian Positional Encodings. All models are similar in size ($\sim 165,000$ parameters) to ensure a fair comparison. For each molecule, we use only its 2D molecular graph, disregarding edge attributes (atom bonding type). Canonical architectures from the literature are employed: Graph Convolutional Networks (Kipf & Welling, 2017), Graph Attention Networks (Veličković et al., 2018), Laplacian Positional Encodings (Dwivedi & Bresson, 2020; Dwivedi et al., 2022), and Transformers (Vaswani et al., 2017). The number of layers is fixed at five but the width of the network is adapted to fit the parameter budget. GAPE are generated using different GNNs pre-trained on various graph alignment datasets - see Table K.10. For each dataset, we select the best architecture trained at the optimal noise level.

*Table K.10.* GNNs used to generate the GAPE, which serves as the positional encoding for the experiments presented in Table K.11. $\eta$ is the noise level used to generate the graph alignment dataset on which we pre-trained the GNN.

| | Base dataset | GNN | Dimension | $\eta$ (training) |
|---|---|---|---|---|
| **GAPE** | (AQSOL) | GatedGCN | 64 | 15% |
| **GAPE** | (PCQM4Mv2) | GatedGCN | 64 | 15% |
| **GAPE** | (Erdös-Rényi) | GAT | 64 | 12% |
| **GAPE** | (OGBN-Arxiv) | GatedGCN | 64 | 18% |

All experiments are repeated ten times, and we report the median validation MAE along with its standard deviation in Table K.11.

These results underscore the effectiveness of our approach, with GAPE achieving the best performance on both datasets. The improvements relative to Laplacian Positional Encodings demonstrate the superior structural information captured by our method. Furthermore, these findings confirm that the Transformer architecture's performance is heavily dependent on the use of effective positional encodings.

MAE varies significantly depending on the model used to generate the positional encodings. Our findings confirm that GAPE generated by a GNN trained on the same dataset as the regression task achieves the best results. Conversely, training the GNN on a dataset with substantially different graph topologies, such as OGBN-Arxiv, leads to suboptimal GAPE and noticeably degrades the transformer's performance. This highlights the importance of carefully selecting the dataset used to train the GNN for generating positional encodings.

Furthermore, this experiment demonstrates that our benchmarking methodology effectively captures useful information that can be leveraged for real-world machine-learning tasks despite being based on a combinatorial task.

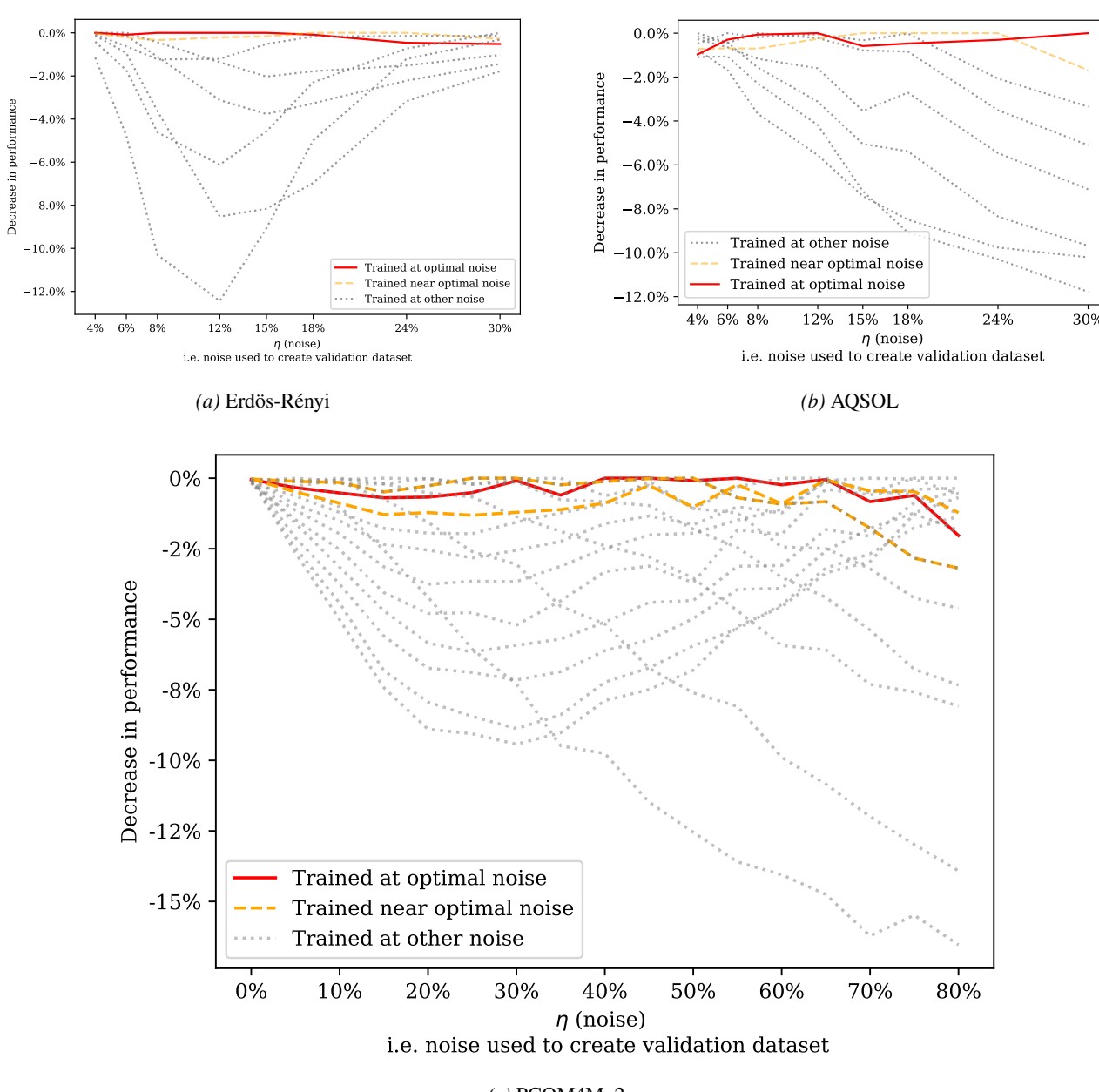

*(a)* Erdös-Rényi

*(b)* AQSOL

*(c)* PCQM4Mv2

*Figure K.2.* Comparison of the generalization error as a function of the noise used to create the validation Graph Alignment dataset. The decrease in performance represents the accuracy gap to the best model.

## L. Experimental Details: GAPE for Molecular Regression Tasks

### L.1. Experimental setup for AQSOL and PCQM4Mv2(subset).

The benchmarked models are trained for 250 epochs on an RTX8000 48GB GPU. Each atom is represented by an embedding vector of dimension 32, obtained with a torch.Embedding layer. For transformers, we sum each atom representation to the corresponding positional encoding. The Adam optimizer is used with a one-cycle learning rate scheduler, which includes 75 warmup steps. The maximum learning rate is set to 0.0004, and a weight decay of 1e-5 is applied to regularize the training.

Due to the size of the PCQM4Mv2 dataset (3.7 million molecules) for the first experiment, we evaluated each model on a subset containing 20,000 molecules in the train split and 2,000 molecules in the validation split.

*Table K.11.* Comparison of GAPE against Laplacian Positional Encodings on the AQSOL dataset and the PCQM4Mv2 (subset) dataset. **Red**: Best performing models

|  | PCQM4Mv2 (MAE) | AQSOL (MAE) |
|---|---|---|
| **GCN** | $0.40 \pm 0.012$ | $1.28 \pm 0.011$ |
| **GAT** | $0.28 \pm 0.008$ | $1.24 \pm 0.011$ |
| **w/o PE** | $0.29 \pm 0.007$ | $1.54 \pm 0.050$ |
| **Laplacian PE** | $0.25 \pm 0.012$ | $1.31 \pm 0.039$ |
| **GAPE (AQSOL)** | $\textcolor{red}{0.17 \pm 0.011}$ | $\textcolor{red}{1.20 \pm 0.036}$ |
| **GAPE (PCQM4Mv2)** | $0.18 \pm 0.007$ | $1.23 \pm 0.026$ |
| **GAPE (Erdös-Rényi)** | $0.18 \pm 0.009$ | $\textcolor{red}{1.20 \pm 0.053}$ |
| **GAPE (OGBN-Arxiv)** | $0.20 \pm 0.017$ | $1.29 \pm 0.052$ |

(Rows "w/o PE" through "GAPE (OGBN-Arxiv)" are grouped under **Transformers**.)

## L.2. Experimental setup for the comparison of GAPE with other types of PE

### L.2.1. DESCRIPTION OF THE MOLECULAR DATASETS

To evaluate the performance of GAPE, we perform three molecular regression tasks:

- The AQSOL dataset (Dwivedi et al., 2022) is a collection of 9,833 molecular graphs, each annotated with aqueous solubility values, the goal is to predict for each molecule the solubility and the performances is compared *via* MAE.

- The PCQM4Mv2 dataset (Hu et al., 2020) is a collection of 3.7 million molecules, the goal is to predict the DFT-calculated HOMO-LUMO energy gap of molecules based on their 2D molecular graphs. The MAE is also used to compare models on this dataset.

- The ZINC dataset (Irwin et al., 2012) is a collection of 249,456 molecules, the goal is to predict the penalized constrained solubility of each molecules based on their 2D molecular graphs. The MAE is also used to compare models on this dataset.

### L.2.2. DESCRIPTION OF THE TRANSFORMER ARCHITECTURE

The results in Section 6.2 are obtained by training a transformer architecture on each molecular dataset. The model uses *torch.nn.Transformer*, with molecular inputs represented as sequences of atom embeddings from *torch.nn.Embedding*. Positional embeddings are added to the atom embeddings before being passed to the transformer. The specific hyperparameters used during training are summarized in Table L.12.

*Table L.12.* Key hyperparameters of the transformer used in Table 1.

|  | AQSOL | ZINC | PCQM4Mv2 |
|---|---|---|---|
| **#Parameters** | 195,218 | 577,953 | 1,217,025 |
| **Layers** | 12 | 12 | 12 |
| **Attention Heads** | 8 | 8 | 16 |
| **Max LR** | 0.0005 | 0.0005 | 0.0001 |
| **Dropout** | 0.1 | 0.1 | 0 |
| **Weight Decay** | 0.0001 | 0.005 | 0 |
| **Batch Size** | 512 | 512 | 1024 |
| **Total Steps** | 10,000 | 150,000 | 200,00 |

As described in Section 6.1, we explained that for generating GAPE it is required to pre-train a GNN on the Graph Alignment task and then use this pretrained model to generating the positional encodings. For optimal performance the graph alignment dataset should be generated at the optimal noise level from a base dataset of similar graph topologies. We summarize the exact models used and Graph Alignment datasets in Table L.13.

*Table L.13.* This table summarizes which GNN model was trained on which Graph Alignment dataset for generating the GAPE used in our experiments.

|  | Base dataset | GNN | Dimension | $\eta$ (training) |
|---|---|---|---|---|
| **AQSOL** | AQSOL | GatedGCN | 32 | 15% |
| **ZINC** | ZINC | GatedGCN | 32 | 30% |
| **PCQM4Mv2** | PCQM4Mv2 | GAT | 32 | 30% |

*Table L.14.* Training times of the different results presented in Section 6.2. We see that pre-training GAPE is not a computational bottleneck as it takes only a fraction of the time needed to train the model.

| Model | AQSOL | ZINC | PCQM4Mv2 |
|---|---|---|---|
| Signet | 1h 15min | 6h | 16h |
| Other models | 30 minutes | 2h | 9h |
| GAPE (Pretraining) | 10 minutes | 1h | 1h |

## M. F1 Curves for Graph reconstruction

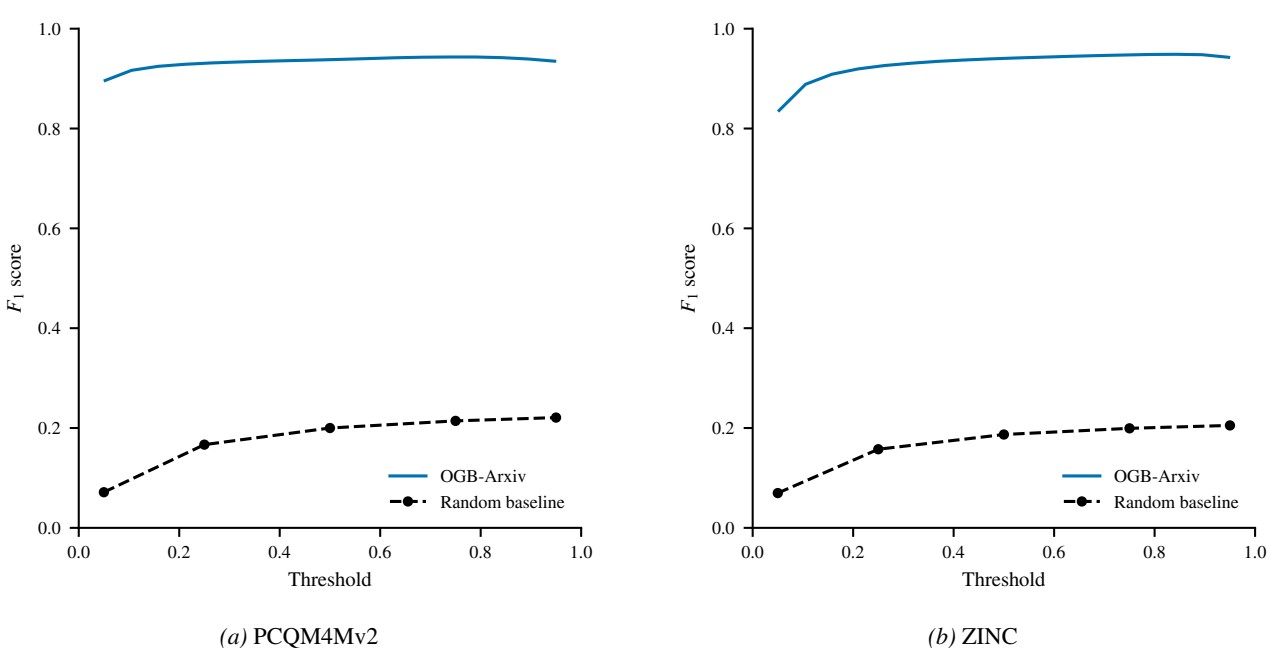

*(a)* PCQM4Mv2                              *(b)* ZINC

*Figure M.3.* $F_1$ score as a function of the classification threshold. Comparison to a random baseline.

## N. Additional experimental results

In Table N.15, we present the results of a new transformer architecture inspired by GraphGPS (Rampášek et al., 2022) to include the edge-features (*i.e.* bond types). This shows promising results for improving the performance of a transformer using GAPE on molecular regression tasks.

*Table N.15.* Result of an advanced Transformer architecture including edge features (*i.e.* with bond types) with Graph Alignment Positional Encodings.

|  | None | RWPE | GAPE | GAPE+RWPE |
|---|---|---|---|---|
| **ZINC (MAE)** | $0.889 \pm 0.0004$ | $0.440 \pm 0.0002$ | $0.0769 \pm 0.0003$ | $0.0395 \pm 0.0002$ |

## O. Statistics of the datasets

*Table O.16.* Key graph statistics for each dataset. Each result is averaged over the whole dataset.

|  | AQSOL | CoraFull | Erdös-Renyi | OGBNArxiv | PCQM4Mv2 | ZINC |
|---|---|---|---|---|---|---|
| **Avg. Degree** | 1.935 | 4.303 | 7.985 | 3.581 | 2.050 | 2.145 |
| **Max. Degree** | 3.219 | 39.925 | 15.367 | 23.138 | 3.199 | 3.275 |
| **Median Degree** | 1.862 | 2.742 | 7.846 | 1.981 | 1.975 | 1.999 |
| **Density** | 0.178 | 0.043 | 0.081 | 0.035 | 0.163 | 0.101 |
| **Diameter** | 8.413 | 5.341 | 4.229 | 6.413 | 7.956 | 12.501 |
| **Radius** | 4.490 | 2.777 | 3.000 | 3.380 | 4.260 | 6.505 |
| **Avg. Clustering** | 0.002 | 0.359 | 0.080 | 0.385 | 0.011 | 0.006 |
| **Transitivity** | 0.002 | 0.190 | 0.080 | 0.320 | 0.011 | 0.006 |
| **Avg. Path Length** | 3.679 | 3.147 | 2.429 | 3.212 | 3.510 | 5.116 |
| **Diameter/Order** | 0.557 | 0.053 | 0.042 | 0.064 | 0.564 | 0.542 |
| **Radius/Order** | 0.304 | 0.028 | 0.030 | 0.034 | 0.303 | 0.282 |

